# Bridging Successor Measure and Online Policy Learning with Flow Matching-Based Representations

**Haosen Shi[1], Jianda Chen[2], Sinno Jialin Pan[1]**
[1]The Chinese University of Hong Kong, [2]Ant International
{haosen.shi.ai@link, sinnopan@}.cuhk.edu.hk jianda.chen@ant-intl.com

## Abstract

The Successor Measure (SM), a powerful method in reinforcement learning (RL), describes discounted future state distributions under a policy, and it has recently been studied using generative modeling techniques. Although SM is a powerful predictive object, it lacks compact representations tailored for online RL. To address this, we introduce *Successor Flow Features* ($\text{SF}^2$), a representation learning framework that bridges SM estimation with policy optimization. $\text{SF}^2$ leverages flow-matching generative models to approximate successor measures, while enforcing a structured linear decomposition into a time-invariant embedding and a time-dependent projection. This yields compact, policy-aware state-action features that integrate readily into standard off-policy algorithms like TD3 and SAC. Experiments on DeepMind Control Suite tasks show that $\text{SF}^2$ improves sample efficiency and training stability compared to strong successor feature baselines. We attribute these gains to the compact representation induced by flow matching, which reduces compounding errors in long-horizon predictions. The code is available on [1].

## 1 Introduction

A key reason for the success of deep reinforcement learning (RL) in complex, sequential real-world tasks is its ability to learn meaningful representations automatically. This is often achieved through neural network architectures with specific inductive bias and efficient representation training algorithms. Effective representations generalize well across different observations and environments, give accurate value estimation, make efficient planning over long decision sequences, and achieve robustness when encountering new observed states (Kulkarni et al., 2016). However, how to find a general and efficient representation learning method that does not rely entirely on reward functions but focuses on environment dynamics is a crucial problem, especially for complex tasks with a continuous state space, sparse reward, and long decision sequences.

Successor Representation (SR) (Dayan, 1993) offers a promising approach by decoupling the reward function from environment dynamics. It captures the expected future state occupancy under a given policy, offering a dual interpretation: it can be viewed as a linear basis for state-action value functions, or equivalently, as a compact representation of infinite-horizon discounted visitation distributions. However, SR is inherently limited to discrete state spaces. To overcome this limitation, Successor Feature (SF) (Barreto et al., 2017) was introduced, incorporating a reward-relevant feature mapping along with a Temporal Difference (TD)-style learning algorithm. A key challenge, however, remains the design of an appropriate feature mapping, which is still an open problem (Ollivier, 2025).

More recently, **Successor Measure** (SM) emerged as a generalization of SR that directly models the discounted future state occupancy. Unlike SF, which relies on predefined features, SM describes distributions in principle in infinite-dimensional space and is typically estimated via generative models. Recent advances such as the *geometric horizon model* (GHM) (Thakoor et al., 2022), the *$\gamma$-model* (Janner et al., 2020), and TDFlow (Farebrother et al., 2025) utilize generative modeling and generalized TD learning to estimate SM, facilitating applications in policy evaluation and transfer

---

[1]https://github.com/Shiien/successor-flow-representation-implementation

learning. In particular, TDFlow builds on recent progress in generative modeling, specifically **flow matching** techniques (Lipman et al., 2023), enabling direct, simulation-free trajectory generation between distributions. This makes it highly efficient for continuous and high-dimensional settings. Furthermore, the mixed nature of flow matching aligns naturally with TD-style training.

The stability and efficiency of flow matching make it exceptionally well-suited for modeling SM, as it mitigates compounding errors over long horizons and enhances scalability in complex environments. However, **online RL** imposes stricter demands, requiring representations that not only retain the predictive power of SM but also adapt quickly to newly observed experiences. These requirements motivate a new framework that unifies the robust long-term forecasting of SM, the stable and efficient generative learning of flow matching, and the rapid adaptability essential for effective online RL. In this paper, we introduce **Successor Flow Features** ($\mathtt{SF}^2$), a new framework that leverages flow matching to approximate SM. $\mathtt{SF}^2$ enforces a structured decomposition into a **time-invariant low-dimensional embedding** of state-action pairs and a **time-dependent projection operator**. This design offers several key advantages: (i) the time-invariant embedding is tailored for online RL, enabling joint training with value functions and seamless integration into existing algorithms like TD3 (Fujimoto et al., 2018) and SAC (Haarnoja et al., 2018); (ii) the time-dependent projection enables generative models to reconstruct SM while decoupling policy-dependent and environmental structure. We evaluate $\mathtt{SF}^2$ by integrating it into TD3 and SAC on continuous-control benchmarks from Mujoco Playground (Zakka et al., 2025). Our results demonstrate improved average performance over standard baselines. While preliminary, these findings indicate that combining flow-based generative modeling with successor representations is a highly promising direction for scalable RL representation learning. Our contributions are threefold:

- We propose a generative model with a linear projection structure to approximate SM via flow matching.
- We introduce $\mathtt{SF}^2$, an informative representation for online RL that is trained jointly with value functions.
- We provide empirical evaluation on challenging continuous-control tasks, together with diagnostic studies of bootstrapped flow matching.

To the best of our knowledge, $\mathtt{SF}^2$ is the first approach to explicitly integrate successor measures with policy optimization for online RL representation learning. We emphasize that $\mathtt{SF}^2$ is an initial step in this direction rather than a complete solution: rigorous theoretical guarantees and broader empirical validation remain important open challenges.

## 2 PRELIMINARY

In this paper, we use uppercase serif fonts to denote a set $\mathsf{S}$, $\mathscr{P}(\mathsf{S})$ to denote the space of probability measures over a measurable set $\mathsf{S}$, uppercase capital letters to denote random variables (e.g., $S$) and $\mathbf{R}^n$ to denote the $n$-dimensional real space.

**Markov Decision Process** We consider a discounted Markov decision process $\mathcal{M} = (\mathsf{S}, \mathsf{A}, P, r, \gamma)$, which includes the state space $\mathsf{S}$, discrete or continuous action space $\mathsf{A}$, transition kernel $P : \mathsf{S} \times \mathsf{A} \to \mathscr{P}(\mathsf{S})$, reward function $r : \mathsf{S} \times \mathsf{A} \to \mathbf{R}$, and discount factor $\gamma \in [0, 1)$. Following the setting of Blier et al. (2021), the state space is measurable (either continuous or discrete). In an MDP, an agent interacts with the environment by observing the current state $s_t \in \mathsf{S}$, selecting an action $a_t \in \mathsf{A}$ according to policy $\pi$, and then receiving a reward $r(s_t, a_t)$ while transitioning to a new state $s_{t+1} \sim P(\cdot|s_t, a_t)$. The objective of reinforcement learning algorithms is to find a policy $\pi : \mathsf{S} \to \mathscr{P}(\mathsf{A})$ that maximizes the expected cumulative reward, or the value function $V^\pi(s) = \mathbb{E}^\pi \left[ \sum_{t=0}^\infty \gamma^t r(s_t, a_t) \mid s_0 = s \right]$ for any state $s \in \mathsf{S}$, where $\mathbb{E}^\pi$ denotes expectation under the distribution induced by policy $\pi$ interacting with the MDP. The value function satisfies the Bellman equation: $V^\pi(s) = \sum_{a \in \mathsf{A}} \pi(a|s) \left[ r(s, a) + \gamma \sum_{s' \in \mathsf{S}} P(s'|s, a) V^\pi(s') \right]$.

**Flow Matching** Flow Matching (FM) (Lipman et al., 2023) is a technique used in generative modeling to learn mappings between distributions. Define a time-dependent diffeomorphic map $\phi_k = \phi(\cdot, k) : \mathbf{R}^n \times [0, 1] \to \mathbf{R}^n$ governed by an Ordinary Differential Equation (ODE) : $\frac{dx_k}{dk} = v(x_k, k)$ with time $k \in [0, 1]$, where $x_k := \phi_k(x_0)$. We use the notation $k$ rather than $t$ for the ODE's time parameter to distinguish it from the timestep $t$ used in MDPs. A Continuous Normalizing Flow, one

kind of Neural Ordinary Differential Equations (Chen et al., 2018), is employed to parameterize the vector field $v_k = v(\cdot, k) : \mathbf{R}^n \times [0,1] \to \mathbf{R}^n$ as $u_\theta(\cdot, k)$ and determines the flow dynamics.

To find the training target of the parameterized time-dependent vector field, FM introduces a mixture representation approach to estimate the marginal vector field as a mixture of conditional vector fields that condition on each data point from $p_{\text{target}}$. For example, given a prior distribution $p_0(x) = \mathcal{N}(0, I_n)$ and a sampled data point $x_1$ from the target distribution $p_1(x) := p_{\text{target}}(x)$, FM constructs conditional probability paths $p_k(x|x_1) = \mathcal{N}(x; \mu_k(x_1), \sigma_k^2(x_1)I_n)$. The corresponding conditional time-dependent diffeomorphic map and the conditional vector field are given by $\phi_k(x, x_1) = \sigma_k(x_1)x + \mu_k(x_1)$ and $\frac{d\phi_k(x, x_1)}{dk}$, respectively. The training objective is

$$\mathcal{L}(\theta) = \mathbb{E}_{\epsilon \sim \mathcal{N}(0,I), x_1 \sim p_{\text{target}}(x), k \sim \mathcal{U}(0,1)} \left[ \left\| u_\theta(\phi_k(\epsilon, x_1), k) - \frac{d\phi_k(\epsilon, x_1)}{dk} \right\|^2 \right],$$

where $\mathcal{U}(0,1)$ represents the uniform distribution over the interval $[0,1]$. This objective minimizes the squared difference between the parameterized vector field and the target vector field at randomly sampled time points, states, and noise values. The time parameter $k$ is uniformly sampled to ensure the model learns the entire trajectory from the prior distribution to the target distribution.

Samples are generated by solving $x_1 = x_0 + \int_0^1 u_\theta(x_k, k)dk$ where $x_0 \sim p_0$ with standard ODE solvers (Gautschi, 2011). When additional conditions are imposed as $c$ for the vector fields $u_\theta(x, k, c)$ and the sampled data conforms to the distribution $x \sim q(\cdot|c)$, the FM framework is also capable of constructing flexible conditional generative models.

**Successor Measure**  The SM (Blier et al., 2021) is a probability distribution over states that captures the expected discounted future state visitations under a given policy, a transition kernel, and a state-action pair. Formally, for a policy $\pi$, the SM $\mu^\pi(\mathsf{X}|s, a)$ represents the probability of visiting state $s' \in \mathsf{X} \subseteq \mathsf{S}$ when starting from state-action pair $(s, a)$ and following policy $\pi$, with geometric discounting:

$$\mu^\pi(\mathsf{X}|s, a) = (1 - \gamma)\mathbb{E}_{(s_1, s_2, \ldots, s_t, \ldots) \sim P^\pi} \left[ \sum_t \gamma^t \mathbb{1}_{s_t \in \mathsf{X}} \mid s_0 = s, a_0 = a \right],$$

where the expectation is taken over all possible trajectories generated by starting at state $s$, taking action $a$, and then following policy $\pi$ for all subsequent steps. The indicator function $\mathbb{1}_{s_t \in \mathsf{X}}$ equals 1 when the state at time $t$ belongs to the set $\mathsf{X}$ and 0 otherwise.

Similar to the Bellman equation for the value function, the SM satisfies the Bellman equation (Blier et al., 2021):

$$\mu^\pi(\mathsf{X}|s, a) = (1 - \gamma)P(\mathsf{X}|s, a) + \gamma \sum_{s' \in \mathcal{S}} P(s'|s, a) \sum_{a' \in \mathcal{A}} \pi(a'|s')\mu^\pi(\mathsf{X}|s', a'). \tag{1}$$

This recursive formulation reveals that the SM can be interpreted as a mixture distribution between the immediate state distribution induced by the transition kernel (with weight $1 - \gamma$) and the bootstrapped future state distribution (with weight $\gamma$). The conditional generative models can be employed to predict the SM utilizing the recursive form equation 1. In general, the learning objective is formulated as a maximum likelihood estimation problem, which aims to find the optimal generative model by solving:

$$\max_\mu \mathbb{E}_{\mathsf{X} \sim (1-\gamma)P(\cdot|s,a) + \gamma\mathbb{E}_{s' \sim P^\pi(\cdot|s,a), a' \sim \pi(\cdot|s')}[\mu^\pi(\cdot|s',a')]}[\log \mu^\pi(\mathsf{X}|s, a)]. \tag{2}$$

While this objective provides a general framework, the specific loss function needs to be adapted according to the choice of generative model. For instance, in the Geometric Horizon Model (GHM) (Thakoor et al., 2022) (also referred to as the $\gamma$-model (Janner et al., 2020)), different implementations employ distinct training losses, such as those based on VAE (Kingma & Welling, 2013) or GAN (Goodfellow et al., 2014). In this work, we focus on the best-performing variant, the flow matching used in TDFlow (Farebrother et al., 2025), which utilizes a modified flow matching loss. We will elaborate on its details in the next section. A further discussion about the benefits of using flow matching for SM through an explicit mixture viewpoint can be found in Appendix A.2.

# 3 SUCCESSOR FLOW FEATURE

## 3.1 FLOW MATCHING FOR SUCCESSOR MEASURE LEARNING

The SM's mixture structure is particularly well-suited for flow matching approaches, allowing us to directly model the interpolation between immediate transitions and future state distributions. For learning $\mu^\pi(s'|s, a)$ with its corresponding parameterized time-dependent vector field $u_\theta(x, k, s, a)$, we utilize its natural mixture representation:

$$\mu^\pi(s'|s, a) = (1 - \gamma)P(s'|s, a) + \gamma\mathbb{E}_{s'' \sim P(\cdot|s,a), a'' \sim \pi(\cdot|s'')}\mu^\pi(s'|s'', a''),$$

which yields the corresponding FM training objective for the parameterized vector field $u_\theta(x, k, s, a)$ on given tuple $(s, a, s')$:

$$\mathcal{L}_{\text{flow}}(\theta) = (1 - \gamma)\mathcal{L}_P(\theta) + \gamma\mathcal{L}_{\text{bootstrapping}}(\theta)$$

$$= (1 - \gamma)\mathbb{E}_{\epsilon, k, s' \sim P(\cdot|s,a)}\left[\left\|u_\theta(\phi_k(\epsilon, s'), k, s, a) - \frac{d\phi_k(\epsilon, s')}{dk}\right\|^2\right]$$

$$+ \gamma\mathbb{E}_{\epsilon, k, a' \sim \pi(\cdot|s'), s_e \sim \mu_\theta(\cdot|s',a')}\left[\left\|u_\theta(\phi_k(\epsilon, s_e), k, s, a) - \frac{d\phi_k(\epsilon, s_e)}{dk}\right\|^2\right], \quad (3)$$

where $\mathcal{L}_P(\theta)$ is used to learn the transition distribution and $\mathcal{L}_{\text{bootstrapping}}(\theta)$ uses a temporal difference form where bootstrapping distributions induced by $u_\theta(\cdot, \cdot, s', a')$ serve as $\mu^\pi(\cdot|s', a')$. However, this sampling procedure introduces substantial computational costs due to the need for multiple network evaluations during the generation process.

To enhance training efficiency, we leverage the fact that sampling from the SM can be achieved through an ODE solver(Euler method as an example): $s_e \sim \mu_\theta(\cdot|s, a)$ is equivalent to $s_e = \text{Euler}(\epsilon, u_\theta(\cdot, \cdot, s, a))$ where $\epsilon \sim \mathcal{N}(0, I_n)$. Motivated by the TD$^2$-CFM loss formulation from (Farebrother et al., 2025), we directly align the vector fields conditioned on different state-action pairs at the same noise level instead of generating full successor states and then comparing them. This approximation avoids expensive ODE integration while preserving the consistency between local flow directions. Intuitively, if two vector fields agree on their evolution, their generated distributions will also agree.

$$\mathcal{L}_{\text{bootstrapping}}(\theta) \approx \mathbb{E}_{\substack{\epsilon, k, a' \sim \pi(\cdot|s') \\ x_k = \text{ODE}(\epsilon, k, u_\theta(\cdot, \cdot, s', a'))}}\left[\|u_\theta(x_k, k, s, a) - u_\theta(x_k, k, s', a')\|^2\right]. \quad (4)$$

This approach aligns the vector fields conditioned on different state-action pairs at the same noise level, eliminating the need to fully generate denoised states and then apply the $\phi_k$ transformation. This form substantially reduces the need for small integration steps in the ODE solver, decreasing computational overhead while maintaining performance. We provide a detailed analysis of the trade-off between computational efficiency and model performance in our ablation studies presented in Section 4.3. It is noted that the final loss used for learning is based on expectation over transition tuples from the current policy $\pi$. We model conditional vector fields on the latent space induced by the flow parameterization (Section 2). This avoids explicit density ratios on S and ensures that training targets remain well-defined even when the SM is provided implicitly via pushforwards.

## 3.2 SUCCESSOR FLOW FEATURE FROM ESTIMATED SUCCESSOR MEASURE

Following the estimation of the SM, we employ a linear projection formulation to derive a compact feature representation.

**Definition 3.1 (Successor Flow Feature)** *We define the Successor Flow Feature ($SF^2$) on state-action pair $(s, a)$ as the output of the mapping $\psi : \mathbf{R}^{dim_S} \times \mathbf{R}^{dim_A} \to \mathbf{R}^d$, which generates the time-dependent conditional vector field $u(s', k, s, a)$ as a linear projection with a time-conditioned matrix field $\zeta : \mathbf{R}^{dim_S} \times [0, 1] \to \mathbf{R}^{d \times dim_S}$:*

$$u(s', k, s, a) = \zeta(s', k)^\top \psi(s, a),$$

*where $\psi(s, a)$ is time-invariant and captures the sufficient dimension reduction property.*

In contrast to conventional conditional generative models that combine conditions, timestamps, and noised inputs through complex non-linear transformations, our approach employs a time-invariant feature $\psi(s, a)$ that interacts only at the final stage with the matrix field $\zeta$. This architectural choice promotes the encoding of temporal structures within $\zeta$ that are essential for effective downstream representation learning. This linear projection approach has been explored in prior work (Shribak et al., 2024), which extracts spectral features from environmental transition dynamics to enhance reinforcement learning performance.

The representation function $\psi(s, a)$ achieves the Sufficient Dimension Reduction (SDR) (Fukumizu et al., 2009) by establishing conditional independence $s_e \perp\!\!\!\perp (s, a) \mid \psi(s, a)$, i.e. $\mu^\pi(s_e|s, a) = \mu^\pi(s_e|\psi(s, a))$, thereby ensuring that the extracted representations comprehensively capture all relevant information about how state-action pairs relate to successor states. Additionally, this formulation exhibits universal approximation properties (Sasaki & Hyvärinen, 2018), enabling it to theoretically approximate any target function to arbitrary accuracy, which makes it particularly effective for modeling SM across diverse policies and environments.

Figure 1: Schematic visualization of how $\psi$, $\zeta$, and $u$ interact. $\psi(s, a)$ encodes the time-invariant flow representation of the current state–action pair. $\zeta_k(s')$ provides a time-varying projection over future states $s'$. Their inner product defines the vector field $u(s', k, s, a)$, which describes how future-state probability mass evolves in the flow-matching objective.

### 3.3 CONNECTION TO SUCCESSOR REPRESENTATION AND DIFFUSION SPECTRAL REPRESENTATION

**Connection to Successor Representation** Let's consider the one-step gradient updating on the parameterized $\psi$ neural network with parameters $\theta$ using equation 3 with equation 4 under a transition tuple $(s, a, s')$ and sampled $k \sim \mathcal{U}(0, 1), \epsilon \sim \mathcal{N}(0, I_n)$. When $k \to 0$, let $\phi_k(\epsilon, x) = kx + (1 - k)\epsilon$, $\frac{d\phi_k(\epsilon, x)}{dk} = x - \epsilon$, we have:

$$\mathcal{L}_{\text{flow}}(\theta) = (1 - \gamma)\left\| \zeta(ks' + (1-k)\epsilon, k)^\top \psi(s, a) - (s' - \epsilon) \right\|^2 + \gamma \left\| \zeta(x_k, k)^\top \psi(s, a) - \zeta(x_k, k)^\top \psi(s', a') \right\|^2.$$

As $k$ approaches 0, we can make the approximation:

$$\zeta(ks' + (1 - k)\epsilon, k) \approx \zeta(\epsilon, 0), \text{ and } \zeta(x_k, k) \approx \zeta(\epsilon, 0).$$

The intermediate point $x_k$ is approximately obtained through a single ODE transformation step:

$$x_k \approx \epsilon + k\zeta(\epsilon, 0)^\top \psi(s', a').$$

Substituting these approximations into the loss function yields the semi-gradient, where we stop the gradient backpropagation on the bootstrapped target. We have

$$\nabla_\theta \mathcal{L} \approx 2 \left[ (1 - \gamma)\left( \zeta(\epsilon, 0)^\top \psi(s, a) - (s' - \epsilon) \right) + \gamma \left( \zeta(\epsilon, 0)^\top \psi(s, a) - \zeta(\epsilon, 0)^\top \psi(s', a') \right) \right] \nabla_\theta \psi(s, a),$$

which can be rewritten more concisely as:

$$\nabla_\theta \mathcal{L} \approx 2 \left[ \psi(s, a)^\top \zeta(\epsilon, 0) - \left( (1 - \gamma)(s' - \epsilon) + \gamma \psi(s', a')^\top \zeta(\epsilon, 0) \right) \right] \nabla_\theta \psi(s, a).$$

This formulation reveals a temporal difference learning structure where the target combines: (1) A direct supervision component $(1 - \gamma)(s' - \epsilon)$ representing immediate information. (2) A discounted bootstrapped component $\gamma \psi(s', a')^\top \zeta(\epsilon, 0)$ that propagates future representations.

Rearranging into a Bellman-like equation, we have

$$\psi(s, a) \leftarrow (1 - \gamma)(\zeta(\epsilon, 0)^T)^+ (s' - \epsilon) + \gamma \psi(s', a'),$$

where $(\cdot)^+$ denotes the Moore-Penrose pseudoinverse. This formulation reveals that our approach learns a Successor Representation with Dayan's definition (Dayan, 1993). In our case, $(1-\gamma)(\zeta(\epsilon,0)^T)^+(s'-\epsilon)$ serves as the basic feature that captures immediate transitions, while the recursive structure $\psi(s,a) = (1-\gamma)(\text{immediate feature}) + \gamma\psi(s',a')$ serves as the bootstrapped part. The process incorporates a novel element where the next state $s'-\epsilon$ undergoes Gaussian noise perturbation before being projected onto the column space defined by $(\zeta(\epsilon,0)^T)^+$. This can be interpreted as learning a basis for the state space that is robust to perturbations, enabling more effective representation of the expected future state occupancy distribution. This yields an SR-like recursion on $\psi$ under fixed $\zeta$. We emphasize this is an approximation to motivate design; we do not claim exact equivalence to SR. The exploration of how the representation behaves and what properties it captures when $k$ takes values significantly away from 0 remains an open question for future investigation.

**Connection to Diffusion Spectral Representation (Shribak et al., 2024)** As $\gamma$ approaches zero, our approach bears resemblance to Diffusion Spectral Representation (Shribak et al., 2024), which employs diffusion models (Song et al., 2021) rather than flow matching and targets transition probabilities instead of SM. The incorporation of future state transitions enables features to encode transition dynamics across extended time horizons. In Section 4, we conduct empirical comparisons between $\text{SF}^2$ and its variant with $\gamma = 0$ to demonstrate that $\text{SF}^2$ achieves better area-under-curve(AUC) performance compared to approaches that focus solely on short-term transition prediction.

### 3.4 PRACTICAL POLICY OPTIMIZATION WITH FLOW SUCCESSOR REPRESENTATION

In this paper, we consider combining the proposed representation learning method with standard online reinforcement learning algorithms on continuous action spaces. We choose SAC (Haarnoja et al., 2018) and TD3 (Fujimoto et al., 2018) as base algorithms. The learned representation is only used for building the state-action value function $Q(\psi_\theta(s,a))$. And the policy will be implicitly influenced through the $\nabla_a Q(\psi_\theta(s,a))$.

To enhance learning stability and performance, we implement two complementary techniques:

**Value Alignment:** We augment the flow-matching objective with a value prediction component:$\mathcal{L}_{\text{total}} = \mathcal{L}_{\text{flow}} + \lambda\mathcal{L}_{\text{value}}$, where $\lambda$ controls the relative weight of value prediction. The value loss follows the standard temporal difference formulation:

$$\mathcal{L}_{\text{value}} = \mathbb{E}_{(s,a,r,s')\sim\mathcal{D}}\left[\left(Q(\psi_\theta(s,a)) - (r + \gamma\max_{a'}Q(\psi_\theta(s',a')))\right)^2\right].$$

This approach is compatible with various RL algorithms and can incorporate techniques such as double Q-learning (Van Hasselt et al., 2016) for improved target estimation.

**Generative Model Smoothing:** We employ exponential moving average (EMA) target networks that update parameters according to $\theta_{\psi'} = (1-\tau)\theta_{\psi'} + \tau\theta_\psi$ and $\theta_{\zeta'} = (1-\tau)\theta_{\zeta'} + \tau\theta_\zeta$ during bootstrapping phases, consistent with established flow matching training practices (Lipman et al., 2023). We perform an ablation analysis of the effectiveness of this moving average coefficient in Section 4.3. Furthermore, the EMA-updated parameters $\theta'_\psi$ also work in the target network for the estimation of the value function, providing an additional layer of stability to learning dynamics. The overall training objective $\mathcal{L}_{\text{total}}$ for learning $\text{SF}^2$, when embedded in off-policy RL, is shown in Algorithm 1. For the TD3-based methods, $y' =$

---

**Algorithm 1** Training $\text{SF}^2$ within Off-Policy RL

1: **Input:** (state, action, next state, next action) tuple $(s,a,s',a')$, networks $(\psi,\zeta,\psi',\zeta')$, and target for value learning $y'$, which depends on the base algorithm
2: Sample $\epsilon \sim \mathcal{N}(0,I)$, $k \sim \mathcal{U}(0,1)$
3: $s_k = k \cdot s' + \epsilon \cdot (1-k)$, $s_{target} = s' - \epsilon$
4: Compute features and next state loss:
5: $\mathcal{L}_{\text{flow}} = \|\psi(s,a)^T\zeta(s_k,k) - s_{target}\|_2^2$
6: Generate state $x$ using numerical integration, start with $x = \epsilon$:
7: $k_{start}, k_{end} = 0, k$
8: $k_{mid} = \frac{1}{2}(k_{start} + k_{end})$
9: $dx = \psi'(s',a')^T\zeta'(x + \frac{1}{2}\psi'(s',a')^T\zeta'(x, k_{start}), k_{mid})$
10: $x = x + (k_{end} - k_{start})dx$
11: Compute generation loss:
12: $\mathcal{L}_{\text{bootstrapping}} = \|\psi(s,a)^T\zeta(x,k) - \psi'(s',a')^T\zeta'(x,k)\|_2^2$
13: $\mathcal{L}_{\text{value}} = (Q(\psi(s,a)) - y')^2$
14: Return $\mathcal{L}_{\text{total}} = (1-\gamma)\mathcal{L}_P + \gamma\mathcal{L}_{\text{bootstrapping}} + \lambda\mathcal{L}_{\text{value}}$

---

$r + \gamma \min(Q'_1, Q'_2)$, where $Q'_1$ and $Q'_2$ are the target Q-networks evaluated at the next state $s'$ and next action $a'$ sampled from the target policy. For the SAC-based method, $y' = r + \gamma \min(Q'_1, Q'_2) - \alpha \log \pi(a'|s')$, where $\alpha$ is the temperature parameter that determines the trade-off between maximizing expected reward and entropy, and is updated according to the original SAC paper (Haarnoja et al., 2018).

## 4    EMPIRICAL EVALUATION

### 4.1    EXPERIMENTAL SETUP

We implemented all experiments using JAX (Bradbury et al., 2018) and Deepmind Haiku (Hennigan et al., 2020) to leverage hardware acceleration. For the DeepMind Control Suite (Tunyasuvunakool et al., 2020), we utilized a GPU-accelerated version, the MuJoCo Playground (Zakka et al., 2025). All algorithms were edited from their respective implementations in the Brax library (Freeman et al., 2021). Each experiment is conducted on a single NVIDIA GeForce RTX 4090 GPU. Detailed architectural specifications, hyperparameter configurations, and environment-specific parameters are provided in Appendix G.

Flow sampling uses Euler integration with 2 function evaluations (NFEs) unless otherwise noted; we sample $k \sim \mathcal{U}(0,1)$ and base noise $\epsilon \sim \mathcal{N}(0, I)$, and condition vector fields on $(s, a)$ and $(s', a')$ as specified in Section 3. We report wall-clock time for representative settings (Section 4.3) and keep baseline parameter counts comparable by reusing encoder widths and depths across methods (Appendix G).

### 4.2    EXPERIMENTS ON CONTINUOUS ACTION SPACES WITH OFF-POLICY LEARNING

We evaluate $\mathrm{SF}^2$ on seven diverse tasks from the DeepMind Control Suite (Tunyasuvunakool et al., 2020), selected to represent a range of challenges in dynamics complexity, reward structure, and control difficulty. We integrate our approach with two commonly used off-policy algorithms: SAC (Haarnoja et al., 2018) and TD3 (Fujimoto et al., 2018). For comparison with the SF method, we also include Chua et al. (2024), a strong SF method designed for the online RL setting. Their work reports substantial improvements over prior SF approaches. To provide a fair and rigorous comparison, we implemented four baseline methods: TD3Sim/SACSim, which closely follows the approach described in the Chua et al. (2024) with the TD3/SAC algorithm, and TD3SimLap/SACSimLap , which removes the Q-function alignment constraint and incorporates an orthogonality objective for feature learning via the graph Laplacian. All methods use identical neural network architectures, the same number of environment steps, and the same number of parallel environments as our proposed approach.

Following the suggestions described in Patterson et al. (2024), we summarize learning dynamics and aggregate performance across seven environments using 15 random seeds for each algorithm variant. For each algorithm variant, we extract evaluation reward curves (mean of episodic returns over evaluation steps using 1000 trajectories) and compute the Area-Under-Curve (AUC). Then the AUC is normalized per environment by linearly scaling to $[0, 1]$ using that environment's minimum and maximum in order to compare across different environments. For each algorithm family (SAC or TD3), the aggregate panels report the median, interquartile mean (IQM; the mean between the 25th and 75th percentiles), mean, and optimality gap (one minus the mean), with 95th percentile confidence intervals computed using 5000 samples over normalized scores. The results in Figure 2 reveal that incorporating $\mathrm{SF}^2$ enhances the performance of TD3 and SAC across most environments over their standard versions, with greater improvements when using the full successor version ($\gamma = 0.99$). Our method consistently outperforms both baseline algorithms and the transition version ($\gamma = 0.0$), demonstrating the importance of incorporating a longer temporal horizon in representation learning. Full per-environment learning curves for both TD3 and SAC variants are deferred to Appendix C.1 (Figure 7).

Interestingly, the performance gains (compared with baseline) are more pronounced for TD3 than for SAC, suggesting that our method may be particularly beneficial for algorithms that struggle with exploration or representation learning, and a deterministic policy may improve the training efficiency for the online successor measure learning. Additionally, the reduced standard deviations in many cases indicate that $\mathrm{SF}^2$ not only improves performance but also enhances stability. Approaches based

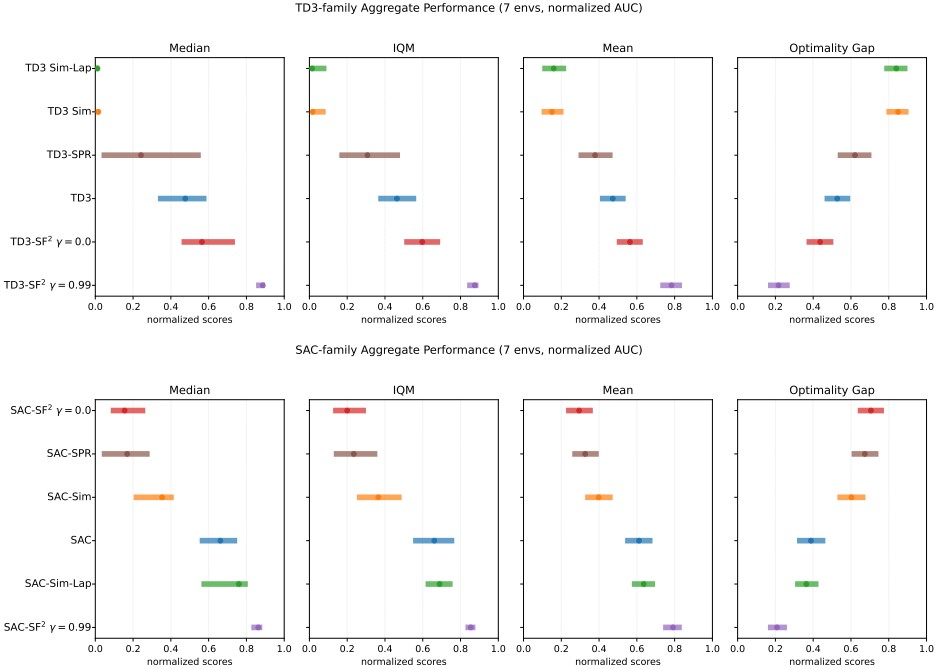

Figure 2: **IQM performance across DeepMind Control tasks.** Panels summarize the aggregate AUC for TD3 (upper) and SAC (lower) variants, comparing vanilla baselines, SF-based baselines, and our $\text{SF}^2$ with transition ($\gamma = 0.0$) and successor ($\gamma = 0.99$) horizons.

on SF tend to struggle on sparse reward tasks based on TD3 experiments, as the majority of transitions yield a reward of 0.0, making it difficult to effectively learn the task weight $w$. In contrast, our method does not depend on this mechanism and thus avoids the associated performance degradation observed in such sparse-reward settings.

## 4.3 HYPERPARAMETER ANALYSIS

We examine the influence of three key hyperparameters in our method on the AcrobotSwingup task: exponential moving average (EMA) coefficient, number of denoising steps, and feature size. We use the mean episode return over the final 50k steps to verify.

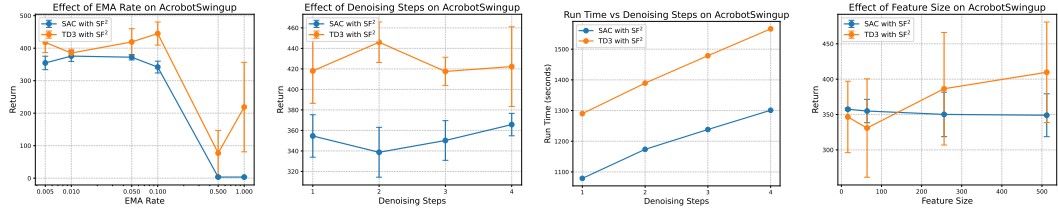

(a) Analysis of exponential moving average coefficient $\tau$ on performance.

(b) Performance evaluation with varying numbers of denoising steps.

(c) Training time evaluation with varying numbers of denoising steps.

(d) Influence of feature size on $\psi$ networks.

Figure 3: Systematic analysis of hyperparameter effects on $\text{SF}^2$ performance. (a) Exponential moving average coefficient $\tau$ demonstrating stability-performance trade-offs, (b) performance sensitivity to denoising step count in the sampling procedure, (c) computational cost scaling with respect to denoising steps, and (d) feature size effects on algorithm performance.

**EMA Parameter.** Figure 3a demonstrates that the EMA coefficient $\tau$ inversely correlates with performance. Peak results occur at $\tau = 0.1$ for TD3 and $\tau = 0.01$ for SAC, with performance

declining as $\tau$ approaches 1.0. Despite considerable variance across configurations, these findings suggest that more stable target network updates (smaller $\tau$ values) enhance learning dynamics in our framework. The results highlight the critical importance of proper $\tau$ calibration, as large values can substantially impair policy effectiveness.

**Denoising Steps.** Our analysis reveals that varying the number of denoising steps produces comparable performance outcomes (Figure 3b), though computational costs increase proportionally with more steps (Figure 3c). We observe that even with minimal denoising steps (1-2), both algorithms maintain robust performance, suggesting that aligning the bootstrapping part can rely on a rough sampling process without requiring extensive iterative refinement. The computational efficiency analysis in Figure 3c further confirms that a small number of denoising steps provides an optimal balance between performance and computational overhead, as the default choice in experiments.

**Feature Size.** As shown in Figure 3d, the two algorithms respond differently to feature size changes. SAC with $\mathtt{SF}^2$ demonstrates similar final 50k steps returns across various feature sizes with low variance, suggesting effective representation learning even in reduced dimensions. TD3 tends to benefit from larger feature sizes, though variance increases, while SAC remains stable across sizes. This indicates that the deterministic policy gradient method particularly benefits from richer feature representations.

## 4.4 COMPUTATION COMPLEXITY

The additional time overhead of our method is due to the additional representation learning and the additional neural network. Here we use the comparison between TD3 and TD3 with $\mathtt{SF}^2$ as an illustration. Each standard TD3 update step consists of one step of critic updating and one step of actor updating, which is consistent with TD3 with $\mathtt{SF}^2$. The main distinction is that an additional feature representation update step: for each update step, TD3 with $\mathtt{SF}^2$ performs seven forward passes and one backward pass through the $\zeta$ network, and two forward passes and one backward pass through the $\psi$ network. Under identical experimental conditions, the original TD3 method's running time on AcrobotSwingup is 659 seconds, while, as reported in Figure 3c, $\mathtt{SF}^2$ with 1 denoising step takes approximately 1300 seconds, about twice as long. Our experiments demonstrate significant improvements in downstream performance, justifying this trade-off. We will further optimize training costs in future work.

## 5 RELATED WORK

**Representation learning in RL.** Reconstruction-based methods have been employed for feature extraction from observations (Hafner et al., 2019; Yarats et al., 2021). Contrastive learning techniques have emerged as a powerful paradigm for learning discriminative state representations (Laskin et al., 2020; Stooke et al., 2021; Zheng et al., 2023). bisimulation metrics offer a more formal approach to learning state abstractions by grouping behaviorally equivalent states (Zhang et al., 2021; Castro et al., 2021; 2023). World models learn to capture the environment's dynamics, allowing agents to plan or learn in a learned latent space (Gelada et al., 2019; Seo et al., 2022; Hafner et al., 2025). Spectral decomposition methods decompose state and actions into low-rank spectral features (Wang et al., 2021b; Yang & Wang, 2020; Shribak et al., 2024). $\mathtt{SF}^2$ uniquely bridges successor measures and online RL: (1) Unlike world models (Gelada et al., 2019; Seo et al., 2022; Hafner et al., 2025) and spectral methods (e.g., (Shribak et al., 2024)) that ignore policy-dependent horizons, $\mathtt{SF}^2$ explicitly encodes discounted future distributions via flow-matched successor measures; (2) While reconstruction methods focus on regenerating the observation, our method considers policy and environment dynamics; (3) Bisimulation methods (Zhang et al., 2021; Castro et al., 2021; 2023) emphasize state similarity with reward, but $\mathtt{SF}^2$ also optimizes features $\psi(s, a)$ from environmental dynamics not only from the value alignment.

**Successor Measure.** SM predicts future state distributions under a given policy, effectively capturing the expected discounted future state occupancy. This concept is closely related to Successor Representations (Dayan, 1993) and Successor Features (Barreto et al., 2017), decoupling environment dynamics from reward structures, facilitating efficient policy evaluation and transfer. Blier et al. (2021) offers a formal mathematical definition of SM and introduces how to estimate it for

value function evaluation. Wiltzer et al. (2024) further enhances these approaches by modeling the full distribution of future state occupancies, providing richer representations for downstream decision-making. GHMs and $\gamma$-models extend the notion of modeling discounted state visitation distributions, creating a continuum between model-free and model-based RL (Thakoor et al., 2022; Janner et al., 2020). Agarwal et al. (2025); Touati & Ollivier (2021) also employ SM to build a representation in the zero-shot RL setting under a precollected offline dataset, always need an extra exploration policy to collect, which allows for optimal policy inference under other given reward functions. $\mathrm{SF}^2$ fundamentally advances this paradigm by introducing the subsequent flow characteristics $\psi(s, a)$ (Definition 3.1) as a structured linear decomposition of the flow field, enabling principled representation learning that greatly expands the scope of application beyond previous methods.

## 6  CONCLUSION

In this work, we proposed the Successor Flow Feature ($\mathrm{SF}^2$) framework, which leverages flow matching and linear-spectral decomposition to address the challenges of estimating and integrating successor measures in online RL. By explicitly modeling the mixture structure of successor measures, our method provides useful state-action representations that facilitate efficient online policy learning and planning. Through extensive empirical evaluations across discrete and continuous control tasks, we demonstrated that using $\mathrm{SF}^2$ consistently improves performance over standard baselines. Our results underscore the promise of flow-based generative modeling for successor features, paving the way for future research on scalable, expressive, and efficient RL representations.

## REPRODUCIBILITY STATEMENT

We facilitate reproducibility by providing an anonymized source-code repository in the supplementary materials. For every experiment, we specify the random seed used, and we document all implementation and training details in Appendix G and D. Together, these references are sufficient for independent researchers to replicate our reported results.

## ACKNOWLEDGMENTS

The research work described in this paper was conducted in the JC STEM Lab of Machine Learning and Symbolic Reasoning funded by The Hong Kong Jockey Club Charities Trust. And we would also like to thank the reviewers for their thorough reading of our manuscript and for their constructive suggestions, which have undoubtedly improved the final version.

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

## A    THEORETICAL FOUNDATION

### A.1    CHALLENGES OF MIXED DISTRIBUTION IN GHMS

The learning procedure for GHMs using generative models, particularly those without explicit density estimation, follows a principled sampling-based strategy maximizing the objective in equation 2. Transition tuples $\{s, a, s'\}$ are sampled from given environment interactions or a pre-collected dataset. For each tuple, an indicator variable is sampled from a Bernoulli distribution with parameter $\gamma$ to determine which distribution to be learned by GHMs: 1) with probability $1 - \gamma$, learn from the true observed successor state $s'$; and 2) with probability $\gamma$, learn from bootstrapped samples, where a state is sampled from the generative model, $\mu(\cdot|s', a')$, conditioned on the observed next state and a next action sampled according to the current policy. This procedure implements the recursive structure of the normalized successor measure (NSM) by sampling.

While theoretically sound, mixture-based learning struggles when $\gamma$ is close to 1 [2]: the data distribution is dominated by the bootstrapped samples from the generative model, introducing significant bias and inaccuracy—especially early in training—with minimal anchoring to real data. In the following section, we demonstrate this challenge via a Gaussian mixture example, where learning the mixed distribution grows increasingly difficult as $\gamma$ approaches unity.

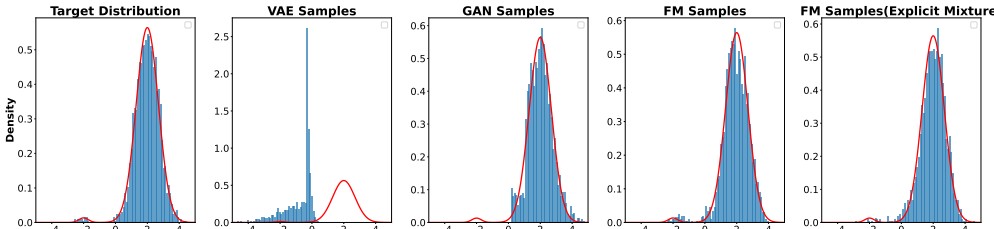

Figure 4: Visualization shows flow-matching methods better capture the multimodal structure of Gaussian mixtures compared to other generative models, such as GANs and VAEs, which is crucial for SM learning with explicit mixture targets.

### A.2    EMPIRICAL VALIDATION ON MIXTURE DISTRIBUTIONS (GAUSSIAN AS AN EXAMPLE)

We evaluate several generative models on a one-dimensional Gaussian mixture that is a simplified yet insightful setting. The target distribution is shown in the left-most part of Figure 4. We train VAEs, GANs, and FM models, and observe that learning becomes difficult as the mixing parameter $\gamma \approx 1$.

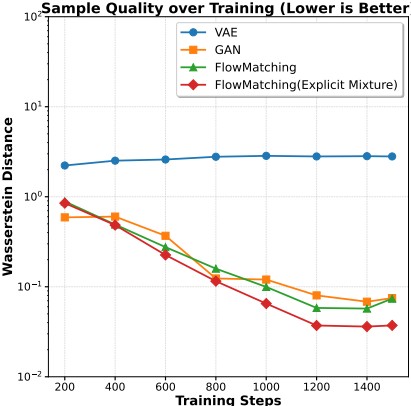

Figure 5: Wasserstein distance between true distribution and generated samples.

Under $\gamma = 0.99$, Figure 4 provides visualizations of the final generated samples, and Figure 5 reports the Wasserstein distance (Vaserstein, 1969) between generated samples and the ground truth

---

[2]To obtain a longer horizon length and align with the discount factor used in the value function definition, the discount factor $\gamma$ is expected to be very close to 1, e.g. $\gamma = 0.99$.

distribution over the training process. These results show how accurately each generative model captures the underlying mixture distribution. Specifically, visualization results show that VAEs and GANs struggle to capture the multimodal structure, while FM exhibits a stronger ability to model the target distribution.

In the next section, we will introduce how to explicitly leverage the mixture structure and corresponding mixture weights to further enhance the FM approach for more efficient and accurate learning of complex multimodal distributions.

### A.3 EXPLICIT MIXTURE OBJECTIVE FOR FLOW MATCHING

To enhance learning of mixture distributions $p^{(\mathrm{mix})}(x) = \gamma p^{(1)}(x) + (1-\gamma)p^{(2)}(x)$, we introduce an **explicit mixture objective** within the flow matching framework. Unlike black-box density estimation, our method directly encodes the known compositional structure of the target distribution into the training objective. This improves computational efficiency and accelerates optimization convergence.

Let $u_\theta(x, k)$ denote the parameterized time-dependent vector field, the marginal distribution on time $k = 0$ is a standard Gaussian distribution, and $\phi_k(\epsilon, x)$ is the conditional time-dependent diffeomorphic map. Let $x \sim p^{(\mathrm{mix})}$, $k \sim \mathcal{U}(0, 1)$, $\epsilon \sim \mathcal{N}(0, I_n)$ and $v_k(\epsilon, x) = \frac{d\phi_k(\epsilon,x)}{dk}$, the original flow matching objective for mixture distributions is,

$$\mathcal{L} = \mathbb{E}_{k,x,\epsilon}\|u_\theta(\phi_k(\epsilon, x), k) - v_k(\epsilon, x)\|^2 = \mathbb{E}_\epsilon\left[\int_0^1\!\!\int p^{(\mathrm{mix})}(x)\|u_\theta(\phi_k(\epsilon, x), k) - v_k(\epsilon, x)\|^2 dxdk\right].$$

Since $p^{\mathrm{mix}}$ is a mixture, we can decompose this into:

$$\mathcal{L} = \mathbb{E}_\epsilon\left[\int_0^1\!\!\int \left(\gamma p^{(1)}(x) + (1-\gamma)p^{(2)}(x)\right)\|u_\theta(\phi_k(\epsilon, x), k) - v_k(\epsilon, x)\|^2 dxdk\right]$$

$$= \gamma\underbrace{\mathbb{E}_{\substack{\epsilon,k \\ x^{(1)}\sim p^{(1)}}}\left\|u_\theta(\phi_k(\epsilon, x^{(1)}), k) - v_k(\epsilon, x^{(1)})\right\|^2}_{\mathcal{L}_1} + (1-\gamma)\underbrace{\mathbb{E}_{\substack{\epsilon,k \\ x^{(2)}\sim p^{(2)}}}\left\|u_\theta(\phi_k(\epsilon, x^{(2)}), k) - v_k(\epsilon, x^{(2)})\right\|^2}_{\mathcal{L}_2}$$

$$= \gamma\mathcal{L}_1 + (1-\gamma)\mathcal{L}_2.$$

This explicit decomposition enables training by sampling from each mixture component and reweighting the loss according to its corresponding mixture weight. By leveraging the known structure of the mixture, this approach aligns closely with the recursive formulation of the NSM, which also exhibits a mixture form rooted in the Bellman equation, which is also noted in (Farebrother et al., 2025). As shown in Figure 4 and Figure 5, our proposed method, *FlowMatching (Explicit Mixture)*, outperforms standard baselines by effectively capturing the multimodal structure of the target distribution. It is worth noting that not only can flow matching methods exploit this form of distribution mixing, but also diffusion models (Song et al., 2021) and bridge-based models (Wang et al., 2021a), which all rely on the diffusion mixture representation (Peluchetti, 2023).

## B EXPERIMENTAL SETUP AND COMPARISONS

### B.1 COMPARISON WITH OTHER REPRESENTATION LEARNING METHODS

To align with our method, we configure SPR (Schwarzer et al., 2021) with $K = 1$, so it relies only on the immediate successor transition, just like our approach. Since the SPR method was originally designed for image input, while our environment uses state input, we use Gaussian noise to simulate data augmentation. In order to select a suitable augmentation setting, we perform a hyperparameter sweep over the Gaussian noise magnitude on the AcrobotSwingup environment.

Following the sensitivity analysis in Figure 6, we adopt the best-performing noise standard deviation of $0.05$ and reuse it across all 7 DeepMind Control Suite environments without further tuning. This configuration is used to generate the aggregate IQM statistics in Figure 2 and the full learning curves in Figure 7. Ensuring that improvements stem from representational benefits rather than environment-specific hyperparameter adjustments. In this configuration, our proposed method exhibits better performance.

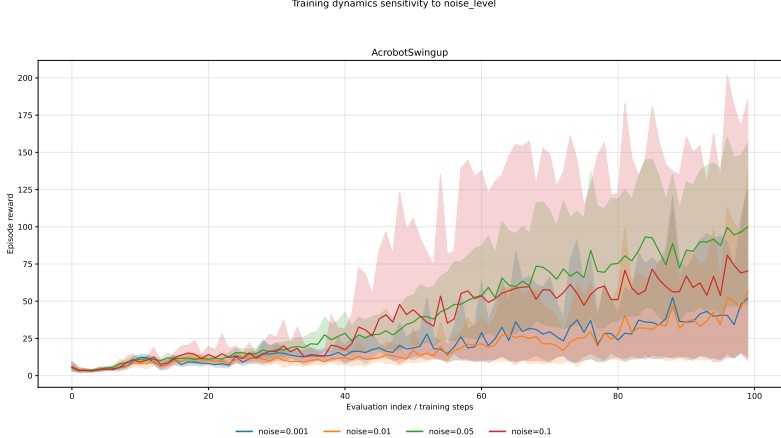

Figure 6: **SPR augmentation sweep.** Performance of SPR on AcrobotSwingup for different Gaussian noise magnitudes used in the augmentation pipeline. The noise level of $0.05$ is selected for the cross-environment experiments.

## C  EXPERIMENTAL RESULTS

### C.1  MAIN RESULTS ON DEFAULT SETTINGS

Figure 7 presents the full learning curves for 7 DeepMind Control Suite tasks using TD3 and SAC variants, complementing the aggregate IQM statistics shown in the main text.

### C.2  ROBUSTNESS ANALYSIS

To rule out the possibility that our gains come only from a specific chosen training budget, we reran every experiment with a 1M-timestep budget and 15 random seeds. Figure 8 reports the IQM aggregates for TD3 and SAC under this stricter setting, showing that $SF^2$ still surpasses vanilla and SF-based baselines by a clear margin. Figure 9 then provides the per-environment learning dynamics, confirming that our method maintains its advantage even when learning is constrained to only 1M timesteps. We also study the effect of gradient steps and generalization to observation noise. Figure 10 (left) shows that allocating more gradient steps to the feature updater steadily improves the IQM score, highlighting that $SF^2$ benefits from thorough feature optimization. For the noise experiment, we train on the 1M-timestep AcrobotSwingup environment using six random seeds, store the checkpoints, and evaluate on perturbed observations with Gaussian noise injected into the observation space. Figure 10 (right) reports the AUC under different noise levels; because $SF^2$ achieves higher performance even before perturbation, it also sustains better AUC after noise injection.

### C.3  EXTENSION TO DISCRETE ACTION SPACES WITH IMAGE INPUT

We conducted additional experiments on the MinAtar benchmark from PGX (Koyamada et al., 2023) using PPO (Schulman et al., 2017). Because our method constructs representations over state-action pairs $(s, a)$, we replaced the standard PPO state-value estimator $V$ with an action-value estimator $Q(s, a)$ and trained it with TD-$\lambda$ target ($\lambda = 0.95$). We also adopted separate actor and critic networks for baseline and our methods. MinAtar provides image-like observations and a discrete action space; to make our approach applicable in this setting, we evaluated two implementations: (i) linear interpolation along image channels (shown as $SF^2$ w/o AE), and (ii) an autoencoder-based pipeline in which the generative model is trained in the latent space (shown as $SF^2$ w AE). All experiments were run with discount factor $\gamma = 0.9$. Results are presented in Figure 11, where we report both IQM (interquartile mean) statistics and training curves. These results provide preliminary evidence that our method can operate effectively in image-based environments with discrete actions.

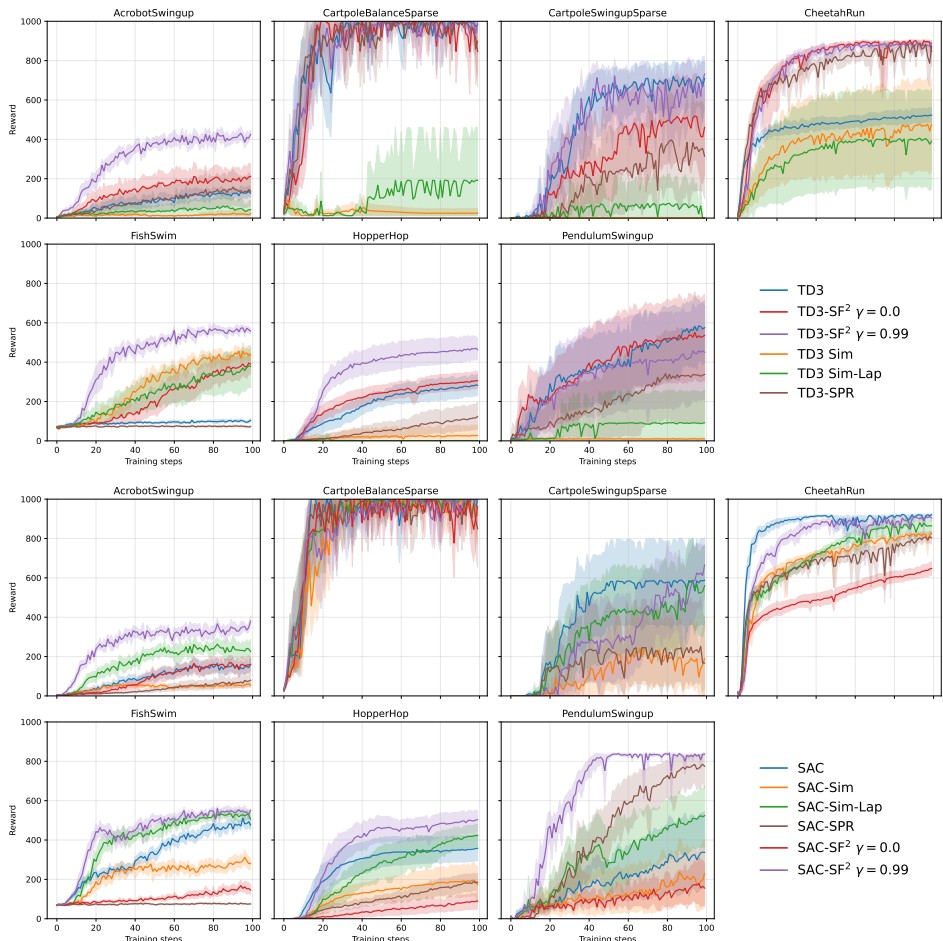

Figure 7: **Learning curves across tasks.** Each plot reports training curve with 95th percentile confidence intervals (bootstrap using 5000 samples) for TD3 (upper) and SAC (lower) with vanilla baselines, SF-based baselines, SPR (Schwarzer et al., 2021), and our $\mathrm{SF}^2$ with transition ($\gamma = 0.0$) and successor ($\gamma = 0.99$) horizons.

## D  DETAILED ALGORITHMS

We demonstrate the detailed algorithm training process of $\mathrm{SF}^2$ when combined with TD3 and SAC, respectively.

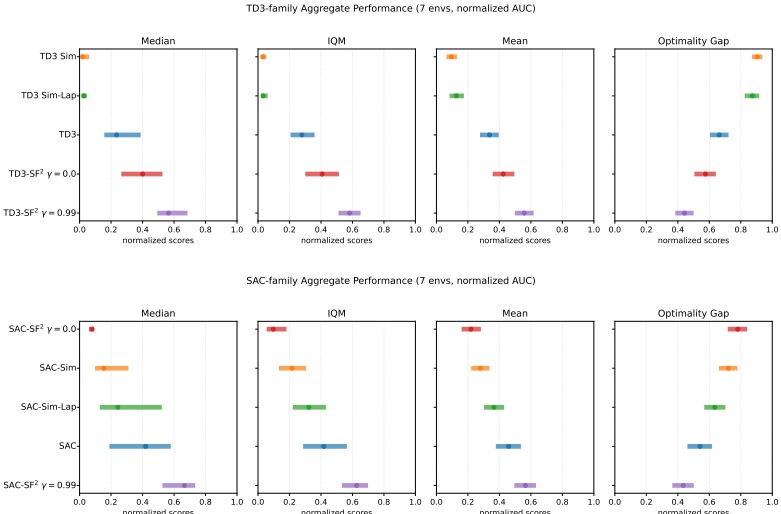

Figure 8: **IQM performance at 1M timesteps.** Aggregate AUC statistics for TD3 (upper) and SAC (lower) at a 1M-step training budget using 15 seeds.

---

**Algorithm 2** TD3 with $\mathrm{SF}^2$

---

1: Initialize $\theta, \phi, \psi, \zeta$; target networks $\theta' \leftarrow \theta, \phi' \leftarrow \phi, \psi' \leftarrow \psi, \zeta' \leftarrow \zeta$; replay buffer $\mathcal{B}$
2: **for** episode = 1 to $M$ **do**
3:     Initialize environment, get initial state $s_0$
4:     **for** step = 1 to $T$ **do**
5:         $a_t = \pi_\theta(s_t) + \epsilon, \epsilon \sim \mathcal{N}(0, \sigma)$; execute $a_t$, observe $r_t, s_{t+1}$; store $(s_t, a_t, r_t, s_{t+1})$ in $\mathcal{B}$
6:         **if** $\mathcal{B}$ is large enough **then**
7:             **for** $G$ gradient steps **do**
8:                 Sample batch $(s, a, r, s')$ from $\mathcal{B}$
9:                 $y = r + \gamma \min_{i=1,2} Q_{\phi'_i}(\psi'(s', \pi_{\theta'}(s')))$
10:                 $\phi_i \leftarrow \phi_i - \alpha_Q \nabla_{\phi_i}(Q_{\phi_i}(\psi(s, a)) - y)^2$
11:                 $\psi \leftarrow \psi - \alpha_\psi \nabla_\psi(Q_{\phi_i}(\psi(s, a)) - y)^2$
12:                 $(\psi, \zeta) \leftarrow (\psi, \zeta) - \alpha_{SR} \nabla_{\psi, \zeta} \mathcal{L}_{SR}$
13:                 **if** step mod $d = 0$ **then**
14:                     $\theta \leftarrow \theta - \alpha_\pi \nabla_\theta Q_{\phi_1}(\psi(s, \pi_\theta(s)))$
15:                     $\theta' \leftarrow \tau\theta + (1 - \tau)\theta'$
16:                     $\phi' \leftarrow \tau\phi + (1 - \tau)\phi'$
17:                     $\psi' \leftarrow \tau\psi + (1 - \tau)\psi'$
18:                     $\zeta' \leftarrow \tau_\zeta \zeta + (1 - \tau_\zeta)\zeta'$
19:                 **end if**
20:             **end for**
21:         **end if**
22:     **end for**
23: **end for**

---

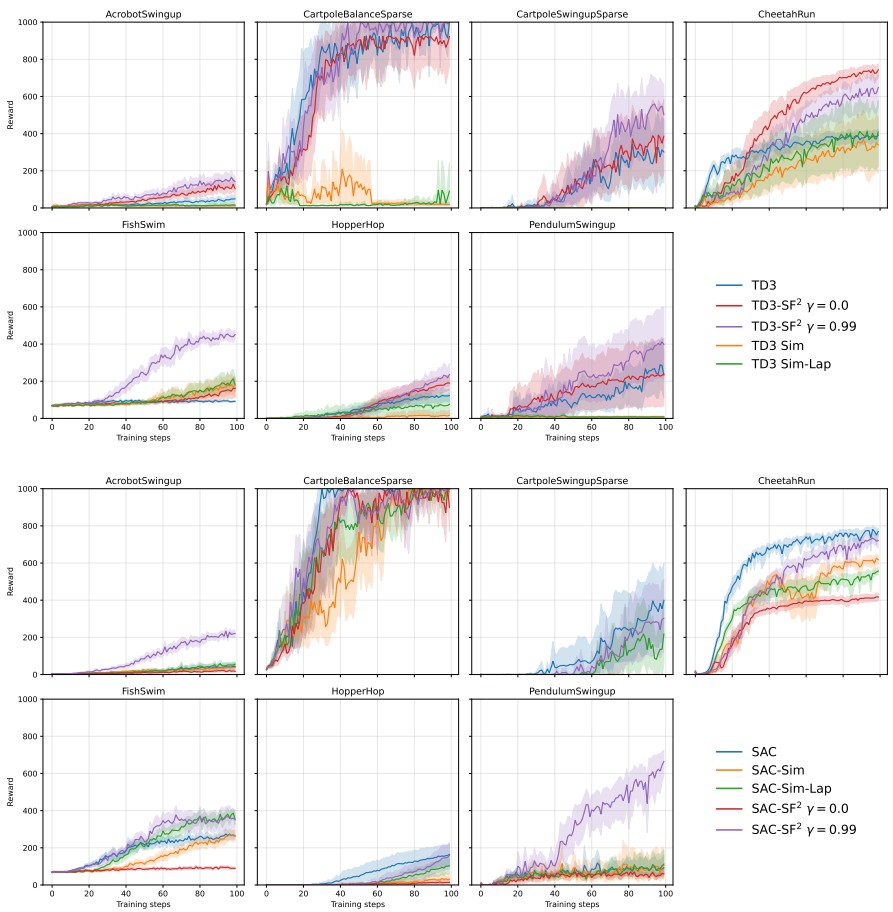

Figure 9: **Learning curves at 1M timesteps.** Each plot shows the median and 95% CIs (bootstrap, 5000 samples) over 15 seeds for TD3 (upper) and SAC (lower) with vanilla baselines, SF-based baselines, SPR (Schwarzer et al., 2021), and our $\texttt{SF}^2$ with transition ($\gamma = 0.0$) and successor ($\gamma = 0.99$) horizons.

---

**Algorithm 3** SAC with $\texttt{SF}^2$

1: Initialize $\theta, \phi, \psi$; target networks $\theta' \leftarrow \theta, \phi' \leftarrow \phi, \psi' \leftarrow \psi$; replay buffer $\mathcal{B}$
2: **for** episode = 1 to $M$ **do**
3:     Initialize environment, get initial state $s_0$
4:     **for** step = 1 to $T$ **do**
5:         $a_t = \pi_\theta(s_t) + \epsilon, \epsilon \sim \mathcal{N}(0, \sigma)$; execute $a_t$, observe $r_t, s_{t+1}$; store $(s_t, a_t, r_t, s_{t+1})$ in $\mathcal{B}$
6:         **if** $\mathcal{B}$ is large enough **then**
7:             **for** $G$ gradient steps **do**
8:                 Sample batch $(s, a, r, s')$ from $\mathcal{B}$
9:                 $y = r + \gamma \min_{i=1,2} Q_{\phi'_i}(\psi'(s', \pi_{\theta'}(s')))$
10:                 $\phi_i \leftarrow \phi_i - \alpha_Q \nabla_{\phi_i}(Q_{\phi_i}(\psi(s, a)) - y)^2$
11:                 $\psi \leftarrow \psi - \alpha_\psi \nabla_\psi(Q_{\phi_i}(\psi(s, a)) - y)^2$
12:                 $(\psi, \zeta) \leftarrow (\psi, \zeta) - \alpha_{SR} \nabla_{\psi, \zeta} \mathcal{L}_{SR}$
13:                 **if** step mod $d = 0$ **then**
14:                     $\theta \leftarrow \theta - \alpha_\pi \nabla_\theta Q_{\phi_1}(\psi(s, \pi_\theta(s)))$
15:                     $\theta' \leftarrow \tau\theta + (1 - \tau)\theta'$
16:                     $\phi' \leftarrow \tau\phi + (1 - \tau)\phi'$
17:                     $\psi' \leftarrow \tau\psi + (1 - \tau)\psi'$
18:                     $\zeta' \leftarrow \tau_\zeta\zeta + (1 - \tau_\zeta)\zeta'$
19:                 **end if**
20:             **end for**
21:         **end if**
22:     **end for**
23: **end for**

---

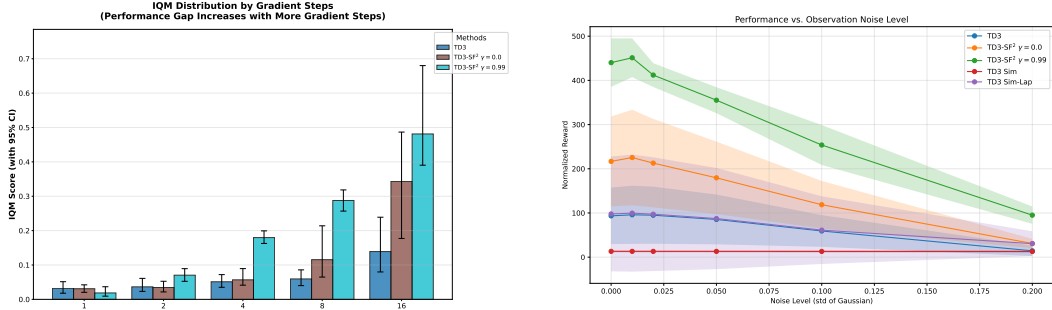

Figure 10: **Left:** Effect of training budget. IQM of the AUC as we vary the number of gradient steps dedicated to representation learning, highlighting the gains from additional feature optimization. **Right:** Generalization under Gaussian observation noise. Interquartile mean performance of policies trained for 1M steps (six seeds) when evaluated on Gaussian-noise-perturbed observations. $\text{SF}^2$ maintains the highest returns.

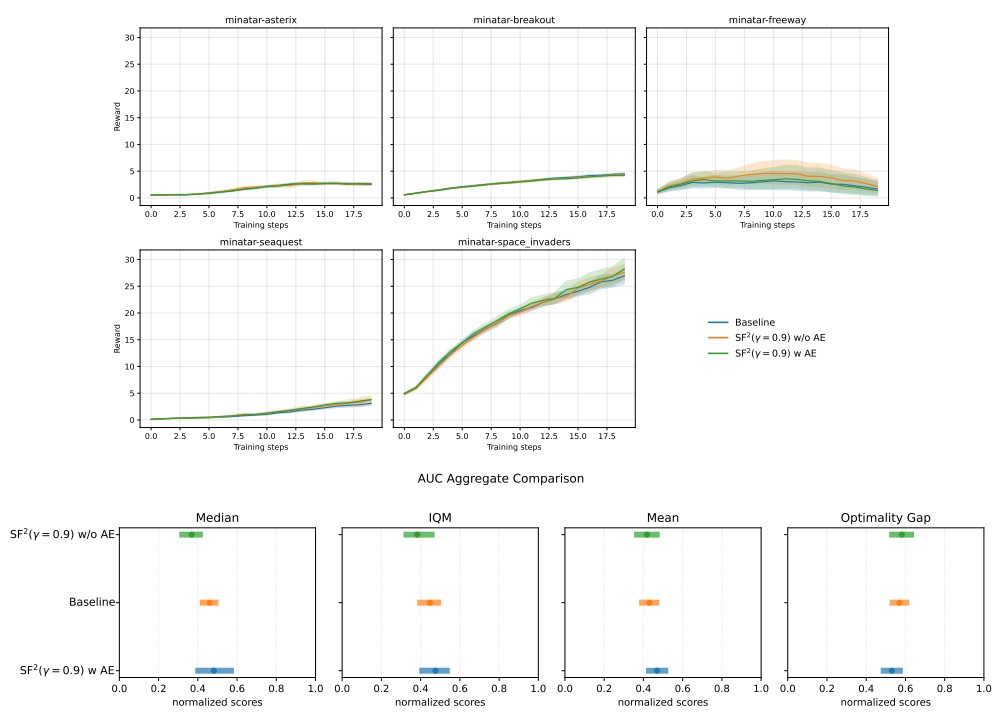

Figure 11: **MinAtar results. Upper:** Learning curves for PPO variants on MinAtar environments with 95th percentile confidence intervals (bootstrap using 5000 samples). **Lower:** Aggregate IQM performance. Interquartile mean AUC statistics across MinAtar environments comparing vanilla PPO, $\text{SF}^2$ without autoencoder (w/o AE), and $\text{SF}^2$ with autoencoder (w AE).

## E  THE USE OF LLMS

The authors used LLMs to polish the language and improve readability. All AI-generated content was thoroughly reviewed and revised by the authors, who take full responsibility for the final content.

## F  EXPERIMENT WITH CRL

We present an additional integration experiment integrating $\text{SF}^2$ into Contrastive RL (CRL) on the Ant locomotion task.

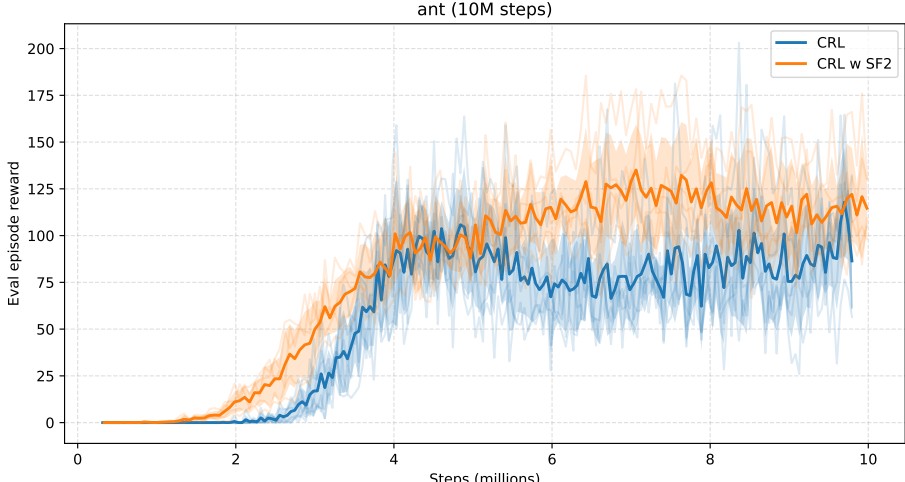

Figure 12: Return curve on Ant environment with CRL (Eysenbach et al., 2022) and CRL with SF$^2$ based on JaxGCRL (Bortkiewicz et al., 2025) (5 seeds).

# G NETWORK ARCHITECTURE AND HYPERPARAMETERS

## G.1 TWO GAUSSIAN EXAMPLE IN SECTION A.2

The target distribution is a one-dimensional bimodal Gaussian mixture with means $\mu_1 = -2.0$ and $\mu_2 = 2.0$, standard deviations $\sigma_1 = 0.3$ and $\sigma_2 = 0.7$, and a mixture weight of $0.01$ (i.e. $\gamma = 0.99$). The generative model is a two-layer MLP with hidden dimension 256 and ReLU activations. Training runs for 1500 steps with batch size 32 and learning rate $10^{-3}$; an exponential moving average of the model weights is maintained with decay rate 0.995. Performance is logged every 100 steps and evaluated every 200 steps. At evaluation time, 4000 samples are generated using 100 numerical integration steps, and the Wasserstein distance to the true distribution is computed.

## G.2 DEEPMIND CONTROL SUITE IN SECTION 4

Table 1: Network and Training Parameters

| Parameter | Value |
| --- | --- |
| **Network Architecture** | |
| Hidden Layer Sizes for Q and policy | (512, 512, 512) |
| Q-Network Layer Normalization | True |
| Policy Network Layer Normalization | True |
| Feature Size | 512 |
| Zeta Network Hidden Layer Sizes | (512, 512) |
| Embedding Size | 64 |
| Activation Function | ReLU |
| Kernel Initializer | LeCun Uniform |
| **Training Parameters** | |
| Number of Timesteps | 5M: CartpoleBalanceSparse, CartpoleSwingupSparse, Fish-Swim; 10M: Acrobot, Hopper, CheetahRun, Pendulum-Swingup |
| Number of Evaluations | 100 |
| Reward Scaling | 1.0 |
| Max Episode Length | 1000 |
| Normalize Observations | True |
| Action Repeat | 1 (4 for PendulumSwingUp) |
| Learning Rate | 1e-3 |
| Number of Environments | 128 |
| Batch Size | 512 |
| Gradient Updates per Step | 8 |
| Max Replay Size | 4,194,304 (1048576 * 4) |
| Min Replay Size | 8192 |
| Discounting Factor | 0.99 |
| Policy Delay | 1 |
| Noise Clip | 0.3 |
| Smoothing Noise | 0.2 |
| Exploration Noise | 0.2 |
| **Optimizer Parameters** | |
| Alpha Optimizer Learning Rate | 3e-4 |
| Policy Optimizer Learning Rate | 1e-4 |
| Q-Network Optimizer Learning Rate | 1e-4 |
| Psi-Zeta Optimizer Learning Rate | 1e-4 |
| **Method-Specific Parameters** | |
| TD3/SAC Gamma for Successor | 0.99 |
| TD3/SAC Tau Zeta | 0.005 |
| TD3/SAC Denoising Steps | 2 Function Evaluations |

