# OpenReview forum: "Bridging Successor Measure and Online Policy Learning with Flow Matching-Based Representations"
_ICLR.cc/2026/Conference — ICLR 2026 Poster_

### Official Review · Reviewer_ydVH · 2025-10-30

**Soundness:** 3
**Presentation:** 3
**Contribution:** 2
**Rating:** 6
**Confidence:** 3

**Summary:**

The paper proposes SF$^2$, a flow-matching based representation that approximates the successor measure (SM) while enforcing a linear decomposition: time-invariant $\psi(s,a)$ and time-dependent $\zeta(\cdot|k)$. SF$^2$ is trained jointly with off-policy RL via a mixture-structured flow matching loss and a value-alignment term, and shows improved average performance and stability on seven DMC tasks, with explicit analysis of EMA, denoising steps, and runtime trade-offs. The work also offers an interpretation that, in a small-$k$ regime, $\psi$ updates resemble succesor representation (SR)-style TD recursion.

---
**Review summary.** The proposed framework introduces a practically effective bridge between SM and flow matching. Despite some theoretical and scalability gaps, the paper offers solid empirical evidence, clear motivation, and transparent ablation studies. With stronger formal grounding and broader experiments, this work could become a notable contribution to representation learning in RL. However, the reviewer feels that the main contribution lies more in practical engineering and integrative design than in fundamental theoretical advancement. Therefore, the reviewer assigns an initial score of 6 and plans to revisit this rating after the authors address the concerns and questions raised in this review.

**Strengths:**

The reviewer thinks that the integration of flow matching and SM is novel and potentially impactful. The method makes the flow-based generative paradigm compatible with online RL, which is a meaningful step beyond prior SM works that focused solely on offline or evaluation settings. Additionally, the connection to diffusion spectral representation is insightful. The discussion of mixture distributions in Appendix A is illuminating, showing that flow matching can better capture multimodal structure than VAEs or GANs (It’s a given, perhaps, anyway), lending theoretical and empirical support to the method’s design.

---
**Writing**
- The paper is written in a clear, structured, and mathematically disciplined manner.
- Equations are well motivated and the derivations are logically consistent. In specific, they provide a clear connection between the mixture structure of the SM and the mixture learning dynamics of Flow Matching.
- This bridige leads to a principled training objective, which is combining direct transition and bootstrap terms weighted by ($1-\gamma$ and $\gamma$).
- The related work section (section 5) is broad and situates SF² well among SR, SM, and flow-based RL.

---
**Methodology**
- The paper presents a creative connection between SM and flow matching. By enforcing a time-invariant $\psi(s,a)$ and time-dependent $\zeta(\cdot|k)$, the method introduces a new linear factorization of dynamics.
- The semi-gradient derivation in section 3.3 gives a clear heuristic connection to SR, showing that $\psi$ updates naturally resemble a TD-style recursion with a flow-matching twist.
- The design promotes a low-dimensional sufficient statistic for state–action pairs, which provide a clear path to sufficient dimension reduction.
- Algorithm 1 and the embedding into SAC/TD3 are modular, allowing the method to be plugged into standard pipelines without major architectural redesign.

---
**Experiments**
- Across all seven DMC tasks, SF2 outperforms or matches baselines, with clear reporting of mean/std, wall-clock training time, and ablations.
- The consideration of $\gamma =0$ and $\gamma = 0.99$ variants shows how long-horizon modeling contributes to performance.
- The reviewer thinks that hyperparameter analyses are informative. For example, the study of EMA rate, denoising steps and feature size. These ablations provides practical insight into stability-efficiency trade-offs.
- In section 4.3 and 4.4, ablations results and computational reports are transparent.

**Weaknesses:**

While the overall empirical presentation is strong, the paper would benefit from a more comprehensive and centralized description of computational details in the main text. Moreover, a clearer breakdown of which components dominate computational load (flow-matching updates vs value alignment vs.actor updates) would make the method’s scalability story much more convincing.

---
**Writing**
- Despite overall clarity, Section 3 contains dense mathematical exposition with several unreferenced variables and occasional overuse of inline equations. Additionally, the reviewer thinks that the interplay between $\psi, \zeta$, and the vector field $u\theta$ could benefit from schematic visualization for intuition.
- Some claims, for example, “SF$^2$ provides compact, expressive, and robust representations,” might seem like overstatements given the limited empirical evidence and absence of stress tests on diverse modalities or noise regimes.

---
**Methodology**
- While the combination of SM and flow matching is powerful, the reviewer thinks that theoretical novelty is somewhat incremental. Each mathmatical part, e.g., flow-mathcing, SM, SDR, has appeared in prior work, so the reviewer thinks that the contribution lies more in enginerring integration than in new theory.
- The derivation in Section 3.3 is heuristic. The claimed connection to SR via the pseudoinverse formulation is intuitive but lacks a formal proof of convergence or representational equivalence. The reviewr thinks that the pseudoinverse handling during training is not described concretely, leaving ambiguity about numerical stability.
- The approximation in Equation (4) that aligns vector fields instead of performing full ODE integration is efficient but somewhat ad-hoc; the implications for consistency and bias in the learned SM are not analyzed theoretically.
- Claims of universal approximation are untested, that is, no experiments validate $\psi$’s robustness under domain shift or adversarial perturbation, which would strengthen the theoretical claims.

---
**Experiments**
- The experiments is limited to state-based DMC tasks. It remains unclear whether the proposed solution scales to high-dimensional observation spaces or more difficult problems, for example, issac gym-based robotics or partially observable environments.
- Comparisons omit several modern flow/diffusion-based RL methods, such as, DSLR, TDFlow, contrastive RL.
- No qualitative analyses, for example, UMAP, t-SNE, or value map, are presented to interpret what $\psi$ or $\zeta$ actually learn.
- Some reported variances are large, especially in P-SwU and C-SwU-S, and the authors do not report statistical significance tests or confidence intervals.

**Questions:**

- Could the authors formalize the small-$k$ connection between $\psi$ updates and SR Bellman recursions? The reviewer wonder that what assumptions does this approximation hold, and what are the limits of this equivalence.
- As the reviewer above-mentioned, some DMC tasks show large performance variance. Could the authors provide the number of seeds and statistical tests (e.g., t-test, bootstrap) to confirm significance?
- The reviewer just wonder how SF2 would extend to pixel-based or image-based input environments. Would $\psi$ and $\zeta$ share encoders or require hierarchical separation?
- Have the authors compared the proposed solutions to recent flow-based RL algorithms?
    - How does the explicit mixture formulation distinguish the proposed solution theoretically or empirically from them?
- How do the authors think that shared encoders or low-rank $\zeta$ to reduce the cost? Is it a good direction or not?

---

> ### Author Response · Authors · 2025-11-21
>
> > Section 3 contains dense mathematical exposition with several unreferenced variables and occasional overuse of inline equation
>
> Would you please point out some instances you found particularly unclear? We would be grateful for pointers and will ensure they are fixed.
>
> > Some claims, for example, “SF provides compact, expressive, and robust representations,” might seem like overstatements given the limited empirical evidence and absence of stress tests on diverse modalities or noise regimes.
>
> We agree to change the language and edit it in the current PDF. We will replace such phrases with  "provides useful representations" due to their higher final performance and higher AUC.
>
> > The derivation in Section 3.3 is heuristic. The claimed connection to SR via the pseudoinverse formulation is intuitive but lacks a formal proof of convergence or representational equivalence.
>
> Section 3.3 is intended as intuition, not an algorithmic step. We do not compute any pseudoinverse during training, hence no numerical stability issue arises.
>
> > The approximation in Equation (4) that aligns vector fields instead of performing full ODE integration is efficient but somewhat ad-hoc; the implications for consistency and bias in the learned SM are not analyzed theoretically.
>
> The alignment objective follows the flow-matching literature, where matching vector fields is a consistent surrogate for pathwise distribution transport under standard regularity conditions.
> To save space, we have not included the proof in our paper; readers can find the corresponding theoretical proof in the TD-Flow paper.
>
> > Claims of universal approximation are untested, that is, no experiments validate’s robustness under domain shift or adversarial perturbation, which would strengthen the theoretical claims
>
> The “universal approximation” statement in our paper is a theoretical assumption that holds in the infinite-dimensional (capacity) limit. Our empirical claims and the effectiveness of the method do not rely on robustness to domain shift or adversarial perturbations, and we do not present universal approximation as a contribution of this work. Robustness under distribution shift or adversaries is valuable future work, but outside the scope of our current study.
>
> > The experiments is limited to state-based DMC tasks. It remains unclear whether the proposed solution scales to high-dimensional observation spaces or more difficult problems, for example, issac gym-based robotics or partially observable environments.
>
> This is a fair limitation. In the real world, there are many scenarios that require multi-step decision-making and use sensor data as input. Furthermore, our method operates in a very similar environment to Isaac gym-based robotics.
> And we note that POMDPs would require recurrent or belief-state encoders, which we leave to future work.
> In addition, we built preliminary results on tasks with the image input and a discrete action space. Please see our reply to Reviewer Cb48.
>
> > Comparisons omit several modern flow/diffusion-based RL methods, such as, DSLR, TDFlow, contrastive RL.
>
> These methods target different settings or components: TDFlow focuses on policy evaluation, while DSLR utilizes a diffusion policy (often in an offline or hybrid RL setting). Most contrastive RL works are not flow/diffusion-based and aim for goal-conditioned RL.
>
> > No qualitative analyses
>
> Because representations play an important role in RL, their quality significantly impacts the performance score. Based on this, we provide performance scores as an indirect indicator of the quality of a representation.
>
> > Could the authors formalize the small-$k$ connection between $\psi$
>  updates and SR Bellman recursions?
>
> Under standard smoothness and boundedness conditions, the update induced by our flow-matching objective is consistent with a discretized SR Bellman recursion as the $k\rightarrow 0$.
>
> > As the reviewer above-mentioned, some DMC tasks show large performance variance. Could the authors provide the number of seeds and statistical tests (e.g., t-test, bootstrap) to confirm significance?
>
> See our reply to Reviewer kuve.
>
> > The reviewer just wonder how SF2 would extend to pixel-based or image-based input environments.
>
> See our reply to Reviewer Cb48.
>
> > Have the authors compared the proposed solutions to recent flow-based RL algorithms?
>
> No, our method is not based on a flow-based RL policy. We are unsure what specific recent literature on flow-based RL algorithms the reviewers mentioned. We would like to know if there is any similar literature that we may have overlooked.
>
> > How do the authors think that shared encoders or low-rank $\zeta$ to reduce the cost? Is it a good direction or not?
>
> Yes. Low-rank adapters and shared early encoders are compatible with our setup and likely to reduce wall-clock time. Thanks for your promising advice.

---

> ### Comment · Reviewer_ydVH · 2025-11-25
>
> > Would you please point out some instances you found particularly unclear? We would be grateful for pointers and will ensure they are fixed.
> - The reviewer thinks that the interplay between $\psi, \zeta$, and the vector field $u\theta$ could benefit from schematic visualization for intuition.
>
> > The alignment objective follows the flow-matching literature, where matching vector fields is a consistent surrogate for pathwise distribution transport under standard regularity conditions. To save space, we have not included the proof in our paper; readers can find the corresponding theoretical proof in the TD-Flow paper.
> - The reviewer was requesting an intuitive explanation of this approximation. In particular, the reviewer would appreciate a brief description of why aligning vector fields serves as a consistent surrogate for full ODE integration in authors’ setting, and what minimal assumptions make this approximation reasonable.
>
> > This is a fair limitation. In the real world, there are many scenarios that require multi-step decision-making and use sensor data as input. Furthermore, our method operates in a very similar environment to Isaac gym-based robotics. And we note that POMDPs would require recurrent or belief-state encoders, which we leave to future work. In addition, we built preliminary results on tasks with the image input and a discrete action space. Please see our reply to Reviewer Cb48.
> - The reviewer cannot agree with the authors’ claim that DMC tasks operate in environments **very similar** to Isaac Gym–based robotics. These two settings differ significantly in observation modality, task complexity, and scalability requirements. Consequently, the reviewer does not consider the current DMC experiments sufficient evidence that the proposed method would scale to Isaac Gym-level environments.
>
> These methods target different settings or components: TDFlow focuses on policy evaluation, while DSLR utilizes a diffusion policy (often in an offline or hybrid RL setting). Most contrastive RL works are not flow/diffusion-based and aim for goal-conditioned RL.
>
> - The reviewer notes that the proposed work is fundamentally a **representation learning method** that happens to employ flow matching, rather than a flow-based policy method. From this perspective, comparisons with contrastive representation methods remain highly relevant and feasible, and the reviewer feels that dismissing such baselines solely on the basis of not being **flow/diffusion-based** is not fully justified.
> - The reviewer notes that the **purpose of improving value learning** is ultimately to obtain a more accurate policy. Given this, it is unclear why the authors dismiss TD-Flow, which also enhances value estimation through a flow-based formulation.
> - Additinoal references for Flow-based RL
>     - Flow Q-learning
>     - Flow‑Based Policy for Online Reinforcement Learning
>     - ReinFlow: Fine-tuning Flow Matching Policy with Online Reinforcement Learning
>     - Flow Matching Policy Gradients
>     - Temporal Difference Flows
>
> Lastly, the reviewer would like to note that considerable effort was devoted to reading, understanding, and evaluating this work in good faith. The reviewer expects a minimal level of appreciation and respect for the time and care invested in this process.

---

> > ### Author Response · Authors · 2025-11-27
> > **Official Comment by Authors [1/2]**
> >
> > We sincerely thank the reviewer for the significant time and effort devoted to carefully reading and evaluating our work. We greatly appreciate the detailed feedback, which has helped us substantially improve the clarity, positioning, and scope of the paper. We will answer your questions point by point.
> >
> > > The reviewer thinks that the interplay between $\psi,\zeta$, and the vector field $u$ could benefit from schematic visualization for intuition.
> >
> > Thanks for your advice. An intuition of our factorization
> > $u_\theta(s',k,s,a)=\zeta(s',k)^\top\psi(s,a)$ is to view it as a standard low-rank decomposition of a multivariate function $f(a,b)$ into two simpler components $g(a)$ and $h(b)$.
> > This type of factorization is widely used in machine learning and offers a powerful inductive bias.
> > In our setting, $\psi(s,a)$ plays the role of a time-invariant,
> > policy-dependent embedding and $\zeta(s',k)$ captures the time-dependent geometry of the flow. Their linear interaction
> > $\zeta(s',k)^\top\psi(s,a)$ yields a flexible yet structured approximation to the true vector field. This inductive bias enhances stability, enables sufficient dimension reduction, and allows the model to capture the major patterns in successor-measure dynamics without requiring a fully nonlinear joint parameterization in all variables. We added a visualization to show the relationship between them in Fig. 11.
> >
> >
> >
> > > The reviewer was requesting an intuitive explanation of this approximation. In particular, the reviewer would appreciate a brief description of why aligning vector fields serves as a consistent surrogate for full ODE integration in authors’ setting, and what minimal assumptions make this approximation reasonable.
> >
> > We thank the reviewer for requesting additional intuition on why aligning vector fields serves as a consistent surrogate for full ODE integration.
> > Our approximation relies on the standard regularity assumptions used
> > throughout the flow-matching and score-based generative modeling literature: Lipschitz continuity of the vector field, uniqueness of the ODE solution, and continuous dependence of the flow on the underlying vector field.
> > Under these assumptions, the global flow is a smooth functional of the local velocity field.
> >
> > The intuition behind this is that if two vector fields agree almost everywhere, then the trajectories induced by integrating these fields must also agree. In other words, matching the infinitesimal motion is sufficient to match the full ODE-defined transport map. This intuition is standard in flow matching and score-based generative models, where learning the local velocity field is known to produce accurate generative flows without explicitly reconstructing entire trajectories during training.
> >
> > For our setting, the successor measure is defined through repeated application of discounted transition dynamics, which is smooth and stable under small perturbations.
> > Thus, learning the vector field that governs the interpolation between the prior and the SM distribution is both sufficient and computationally more efficient than generating full successor samples at every update. Fig.~2b in the original paper. Performance remains stable with different Euler steps, confirming that accurate vector-field alignment captures the relevant structure, and full ODE integration is unnecessary during training.
> >
> >
> > > The reviewer cannot agree with the authors’ claim that DMC tasks operate in environments very similar to Isaac Gym–based robotics. These two settings differ significantly in observation modality, task complexity, and scalability requirements. Consequently, the reviewer does not consider the current DMC experiments sufficient evidence that the proposed method would scale to Isaac Gym-level environments.
> >
> > We thank the reviewer for the clarification. Our intention was not to claim that DeepMind Control Suite (DMC) and Isaac Gym are equivalent. We fully agree that Isaac Gym involves richer sensing pipelines and substantially larger-scale parallelization.
> >
> > However, from the perspective of reinforcement learning algorithm development, the two frameworks do share several structural similarities that motivated our experimental setup. Both DMC and Isaac Gym implement continuous-control robotic tasks with comparable articulated-body dynamics, both expose low-level state-based observations (joint positions, velocities, actuator states), and both support GPU-accelerated simulation. From the standpoint of studying representation learning for continuous control, these shared properties make DMC a reasonable and widely adopted first testbed before moving to large-scale robotics simulators.
> > We agree that demonstrating scalability to Isaac Gym is an important direction. A full evaluation of Isaac Gym–level tasks is an exciting avenue for future work.

---

> > ### Author Response · Authors · 2025-11-27
> > **Official Comment by Authors [2/2]**
> >
> > > The reviewer notes that the proposed work is fundamentally a representation learning method that happens to employ flow matching, rather than a flow-based policy method. From this perspective, comparisons with contrastive representation methods remain highly relevant and feasible, and the reviewer feels that dismissing such baselines solely on the basis of not being flow/diffusion-based is not fully justified.
> > > The reviewer notes that the purpose of improving value learning is ultimately to obtain a more accurate policy. Given this, it is unclear why the authors dismiss TD-Flow, which also enhances value estimation through a flow-based formulation.
> >
> > We thank the reviewer for raising this point. Most CRL methods are designed for the goal-conditioned RL setting, where representations are explicitly trained to support goal-reaching or hindsight relabeling. Extending CRL methods to the general online RL setting without goal specifications is a non-trivial open problem, and existing algorithms do not directly apply to our setting. For this reason, we believe a full CRL comparison is beyond the scope of a rebuttal-stage revision.
> >
> > Nevertheless, to support the reviewer’s suggestion, we conducted an additional experiment where we augmented a representative CRL method with our auxiliary task. The learning curves (Fig. 12 in the updated appendix) show that incorporating SF$^2$ improves or matches the performance of CRL, suggesting that our representation is complementary rather than incompatible.
> >
> > We thank the reviewer for highlighting the related literature on flow-based reinforcement learning. We would like to clarify that the objectives of these works differ fundamentally from SF$^2$, and therefore, a direct comparison would not be meaningful within the scope of this submission.
> >
> > The majority of flow-based RL algorithms cited by the reviewer focus on **policy parameterization**---that is, modeling the **action distribution** using normalizing flows (e.g., flow-based Q-learning, flow-based policy gradients, ReinFlow). These approaches replace the Gaussian policy in SAC or the deterministic policy in TD3 with a flow-based action generator. In contrast, SF$^2$ uses flow matching solely in the **state space** to estimate successor-measure-based representations. Our contribution is orthogonal to policy parameterization, and choosing different policy classes does not affect the nature of the representation-learning problem we study.
> >
> > TD-Flow specifically addresses policy evaluation, optimizing the **MSE** between generated Q-values and ground-truth Q-values in offline regimes. Our setting is standard online RL, evaluated by **return maximization**, and our method is trained jointly with the policy through bootstrapped temporal difference updates. Because the goals, metrics, and experimental setups are substantially different, aligning the two methods for comparison during rebuttal would not be appropriate. SF$^2$ is complementary to these methods, as it can be paired with any policy class---Gaussian, deterministic, or flow-based---and focuses specifically on learning compact successor-driven representations for online RL.
> >
> > Nevertheless, to support the reviewer’s suggestion, we conducted an additional experiment on Flow Matching Policy Gradients(FPO). The reason for choosing this paper is that Flow Q - Learning is devised for offline RL. Additionally, ReinFlow necessitates a pre-trained flow model, and the Flow-based Policy for Online Reinforcement Learning has not yet released its Jax code.
> > However, FPO uses PPO as the base RL method, and our SF$^2$ method needs a Q function. We change the baseline estimation from standard GAE(using $r+\gamma V' - V$) to $Q(\lambda)$. Due to limited time, we didn't get a good performance on this setting. The result is shown in Fig. 13 in the Appendix.
> >
> >
> >
> > We fully acknowledge the reviewer’s concerns and have made extensive revisions to address them. We are grateful for the constructive criticism and for the opportunity to further strengthen our submission. Accordingly, we are making available the code of FPO and CRL for your review. The link is https://anonymous.4open.science/r/only_for_iclr_rebuttal_supply_exp-6AF6/ .

---

### Official Review · Reviewer_kuve · 2025-10-31

**Soundness:** 1
**Presentation:** 2
**Contribution:** 2
**Rating:** 2
**Confidence:** 5

**Summary:**

In RL, the successor measure is the probability mass of ending up in a set of states conditioned on starting in some state, taking an action, and then following some policy. It generalizes the well-known idea of the successor representation to continuous state spaces. This paper introduces a method for estimating the successor measure using the generative modelling technique of flow matching. This method is then used as an auxiliary task to shape an online RL agent’s representation. Empirical evaluation on a set of continuous control tasks show that the method increases the final expected return achieved by the agent.

**Strengths:**

- While prior work appears to have studied how to estimate the successor measure, this work appears to be the first to do so for model-free, online RL agents.
- While I didn’t fully appreciate how the SM estimation part of the algorithm is distinct from prior work, I thought it was interesting how the work integrates a boostrap objective with an generative modelling technique.

**Weaknesses:**

- I’m not convinced that the experiments substantiate the claims made in the paper. Experiments are only repeated for 5 trials and the paper reports large standard deviations. Given the spread and limited trials, it’s plausible that the novel method does not improve upon the simpler base algorithm (SAC or TD3). Please see “Empirical Design in Reinforcement Learning” for a great reference on why these details matter.
- The paper contains a lot of vague, overclaiming language: “achieves superior performance” (superior in what respect?) “remarkable stability” (what is remarkable about it).
- After going through the experiments in detail and ignoring the statistical issues, I think the best claim would be “our method had the largest final expected return achieved when we stopped training.” While final performance obtained is a valid objective to study, it would be valuable to understand how the termination of training was selected. In the appendix, it is mentioned that training went for 10 million timesteps. This is substantially longer than what is usually used on thest environments (1 million seems more common). Why use the longer time? And, in any case, how can we be sure that performance wouldn’t be different for different cut-off points?
- Abstract claims sample efficiency improves — I don’t see any result (e.g., learning curve) substantiating this claim as only final numbers are given.

**Questions:**

I don't fully understand how the SM learning auxilliary task differs significantly from prior work on SM learning. I understand the novelty comes from combining with online model-free RL. In addition to responding to the weaknesses raised above, I'd appreciate gaining some insight into the novelty of learning the SM.

---

> ### Author Response · Authors · 2025-11-21
>
> Dear Reviewer kuve,
>
> Thank you for taking the time to carefully review our paper.
> We hope that the additional experimental results in the **revised version** of the manuscript will alleviate your concerns about the experimental aspects of our method. We greatly appreciate the provision of a clear and feasible reference. We have enhanced our experiments based on your suggestions.
>
> > I’m not convinced that the experiments substantiate the claims made in the paper. ... for a great reference on why these details matter.
>
> We appreciate the reviewer’s concerns and have revised both our experiments and writings. We added the results of AUC(0-T) over 15 random seeds(in the main body, Figure 1), provided the training curve(Figure 6), additional results with 1M environment steps(summarized in Figure 7 and training curves in Figure 8), and additional results on TD3 variants with different gradient steps(Figure 9, left).
>
> > The paper contains a lot of vague, overclaiming language: “achieves superior performance” (superior in what respect?) “remarkable stability” (what is remarkable about it).
>
> Thank you very much for pointing out these points. We agree to revise the wording to make the description more accurate and clear. We change "achieves superior performance" to "better area-under-curve(AUC)" and "remarkable stability" to "similar final 50k steps returns".
>
> > After going through the experiments in detail and ignoring the statistical issues, ..., how can we be sure that performance wouldn’t be different for different cut-off points?
>
> We do not manually determine the training steps ourselves. Instead, we use the **default** number of steps specified for the SAC method in MuJoCo Playground.
> To rule out the possibility that our gains come only from a specific chosen training budget, we reran every experiment with a 1M-timestep budget and 15 random seeds with 16 grads per step compared with 8 in the default setting.
> We summarize the median, mean, and IQM of AUC(0-T) in Figure 7 and the training curve in Figure 8. These results confirm that our method maintains its advantage over the baselines. In addition, we study how our approach benefits from more training budgets and more training times because we require additional training on both the generative and representation models.
> To further illustrate this, we plot the IQM of the AUC as we vary the number of gradient steps. Figure 9(left) shows that allocating more gradient steps for each sample steadily improves the IQM score, highlighting that our method benefits from thorough feature optimization rather than relying on a particular environment budget.
>
> > Abstract claims sample efficiency improves — I don’t see any result (e.g., learning curve) substantiating this claim as only final numbers are given.
>
> Thanks for your advice. We provided the training curve in Figure 6. Each plot reports the training curve with the 95th percentile
> confidence intervals(bootstrap using 5000 samples) for TD3 (upper) and SAC (lower) with vanilla baselines, SF-based baselines, SPR, and our methods with different $\gamma$s.
>
>
> > I don't fully understand how the SM learning auxilliary task differs significantly from prior work on SM learning. I understand the novelty comes from combining with online model-free RL. In addition to responding to the weaknesses raised above, I'd appreciate gaining some insight into the novelty of learning the SM.
>
>
> We are not trivially combining SM learning as an auxiliary task, but proposing a method for extracting the representation from SM. This is still a core and under-explored research problem.
> Our insight is that a well-trained generation model for successor measures must contain long-term information about the current policy and the environment. Recent developments, such as TDflow, have successfully modeled successor measures without relying on any pre-defined abstract representations. This is a chance to exploit such information from a model into representation learning. Compared to single-step transitions, successor measures are a longer-horizon model of the environment that incorporates long-term information. Our **core contribution** is to extract this information to aid representation learning under large policy change(online RL setting) while exhibiting benefit. This is a new attempt, and we successfully made it practical in an online reinforcement learning environment, so we believe our insights can help future research to some extent.
>
> ---
>
> If any concerns remain, we are happy to continue the discussion!

---

> ### Author Response · Authors · 2025-11-27
>
> Dear reviewer kuve,
>
> I hope this message finds you well. As the discussion period is approaching its deadline, we would like to kindly ask whether our rebuttal has addressed your concerns satisfactorily.If there are any additional points or clarifications you would like us to provide, please feel free to let us know. Your insights are invaluable to us, and we're eager to address any remaining issues to improve our work.
>
> Thank you for your careful review.
>
> Best,
>
> The Authors

---

### Official Review · Reviewer_Cb48 · 2025-11-01

**Soundness:** 3
**Presentation:** 3
**Contribution:** 2
**Rating:** 4
**Confidence:** 2

**Summary:**

This paper proposes Successor Flow Features (SF²), a representation learning framework that integrates flow matching with Successor Measure (SM) to bridge SM estimation and online policy optimization, to enhance sample efficiency of RL training. SF² enforces a structured linear decomposition into a time-invariant embedding and a time-dependent projection, enabling seamless integration with off-policy algorithms like TD3 and SAC. Empirical results on DeepMind Control Suite tasks show improved sample efficiency and training stability over successor feature baselines. The core idea of combining flow-based generative modeling with successor representations is theoretically novel, but several critical gaps limit its comprehensiveness and practical impact.

**Strengths:**

1. The integration of flow matching with SM for online RL is a creative extension of both generative modeling and successor representation paradigms. The structured linear decomposition (time-invariant embedding + time-dependent projection) addresses SM’s lack of compact, online-friendly representations, which is a non-trivial technical contribution.
2. Experiments on 7 DeepMind Control Suite tasks demonstrate consistent performance gains over standard TD3/SAC and SF-based baselines. The detailed hyperparameter analysis (EMA coefficient, denoising steps, feature size) and ablation studies add rigor to the results.
3. The paper establishes meaningful links between SF² and existing methods (Successor Representation, Diffusion Spectral Representation), providing conceptual clarity and grounding the framework in prior literature.

**Weaknesses:**

1. Insufficient Justification for Successor Representations in Online RL: The paper frames sample efficiency as a key advantage, but successor representations are inherently designed for reward-dynamics decoupling (enabling generalization/zero-shot transfer). No evidence is provided that SF² outperforms state-of-the-art sample-efficient representation methods (e.g., SPR, CURL) that do not rely on successor paradigms. This raises questions about whether successor representations are necessary for online RL’s core goal of sample-efficient single-task learning.
2. Missing Comparisons with Critical Baselines: The baselines are limited to standard TD3/SAC and SF variants (TD3Sim/SACSim). The absence of comparisons with representative sample-efficient RL methods—particularly SPR (Self-Predictive Representations), which directly targets data efficiency via self-supervised representations—undermines claims of advancing the state-of-the-art in online RL.
3. Unproven Generalization to Pixel-Based Environments: All experiments use structured state features (e.g., joint angles, velocities) from DeepMind Control Suite. The paper provides no analysis or modifications for pixel-based inputs (high-dimensional, unstructured data), which are common in real-world online RL. It remains unclear if SF²’s linear decomposition and flow matching components can adapt to image data.
4. No Evaluation on Discrete Action Spaces: The work exclusively focuses on continuous control tasks with TD3/SAC. Successor representations (e.g., SR) originated in discrete state-action spaces, but SF²’s performance and adaptability to discrete actions are unaddressed. This limits the framework’s generality, as discrete action scenarios (e.g., game playing, recommendation systems) are central to online RL.
5. Complexity: The paper notes SF²’s ~2x longer running time than standard TD3 (e.g., 1300s vs. 659s on AcrobotSwingup).

Reference

SPR: Schwarzer, Max, et al. "Data-Efficient Reinforcement Learning with Self-Predictive Representations." ICLR 2021.

CURL: Laskin, Michael, Aravind Srinivas, and Pieter Abbeel. "Curl: Contrastive unsupervised representations for reinforcement learning." ICML 2020.

**Questions:**

1. Does SF² bring better generalization performance?
2. In what RL scenarios is SF² most effective?

---

> ### Author Response · Authors · 2025-11-21
>
> Dear Reviewer Cb48,
>
> Thank you for taking the time to review our paper. Please see the response below and our edited manuscript.
>
> > Insufficient Justification for Successor Representations in Online RL and Missing Comparisons with Critical Baselines
>
> Thanks for your advice. We added an additional baseline to solve your concern(SPR, which is later published than CURL and shows a better performance).
> To align with our method, we configure SPR with $K=1$, so it relies only on the immediate successor transition, just like our approach.
> Since the SPR method was originally designed for image input, while our environment uses state input, we use Gaussian noise to simulate data augmentation.
> In order to select a suitable augmentation setting, we perform a hyperparameter sweep over the Gaussian noise magnitude on AcrobotSwingup environment. The result is shown in Figure 5, where $\sigma=0.05$ is the best augmentation setting. This setting was then fixed across all environments. Aggregate results are reported in Figure 1, and the corresponding training curves are provided in Figure 6(15 random seeds). Overall, our method has a better AUC performance under this setting.
>
> > Unproven Generalization to Pixel-Based Environments
>
> We conducted additional experiments with PPO on the MinAtar benchmark, which consists of high-dimensional image-like inputs with a discrete action space. Because our method constructs representations over state–action pairs $(s,a)$, we replaced the standard PPO state-value estimator
> $V$ with an action-value estimator $Q(s,a)$ and trained it with TD–$\lambda$ target($\lambda=0.95$).
> We also adopted separate actor and critic networks for baseline and our methods. MinAtar provides image-like observations and a discrete action space; to make our approach applicable in this setting, we evaluated two implementations: (i) linear interpolation along image channels(shown as SF$^2$ w/o AE), and (ii) an autoencoder-based pipeline in which the generative model is trained in the latent space(shown as SF$^2$ w AE). For our method, we use SF$^2$ with discount factor $\gamma=0.9$. Results are presented in Figure 10, where we report both IQM (interquartile mean) statistics and training curves, in Figure 10. These results provide **preliminary** evidence that our method can operate effectively in image-based environments with discrete actions.
>
> > Complexity.
>
> We acknowledge that SF$^2$ incurs a higher per-step wall-clock cost (e.g., ~2× TD3 on AcrobotSwingup) due to the generation model part. However, we have a better AUC performance under the same sample budget. And our method can benefit more from more gradient updates, see our reply to Reviewer kuve.
>
> > Does SF² bring better generalization performance?
>
>
> Regarding generalization, we provide a robustness evaluation with observation noise (see Figure 9, right). For each method, we saved policies from five random seeds; at each noise level, each saved policy was evaluated over 1,000 randomized rollouts, and we report the mean performance. Our method maintains strong returns across noise levels. We attribute this primarily to the stronger base policy obtained at training time rather than to an explicit generalization mechanism. Methodologically, our representation is applied through the critic (the Q-function) and does not directly condition the policy network; the policy benefits indirectly via the value gradients used in TD3/SAC-style updates. Moreover, SF$^2$ was not explicitly designed to optimize representation generalization; its objective emphasizes extracting useful information from the successor measure (i.e., transition dynamics) rather than shaping state features for robustness. Consequently, while the noise experiments show encouraging robustness, we do not claim inherent superiority in generalization as a core contribution of this work.
>
> > In what RL scenarios is SF² most effective?
>
>
> SF$^2$ models the successor measure, i.e., the discounted distribution of future states under the current policy, thereby extracting information about the transition dynamics that is directly useful for value estimation. Consequently, we expect SF$^2$ to be most effective in RL settings where multi-step credit assignment and long-horizon dynamics dominate performance. We select long-horizon continuous control tasks(with 1000 length per trajectory) with delayed or sparsely shaped rewards in our paper, where accurate prediction of downstream state distributions stabilizes critic learning and accelerates policy improvement, where improved critic targets translate into better actor updates through $\nabla_a Q$.
> And in order to achieve enough gradient updates, we employ online off-policy training regimes.

---

> > ### Comment · Reviewer_Cb48 · 2025-11-23
> >
> > I appreciate the detailed supplementary experiments made by authors. Some of my concerns have been addressed. I am still curious about why SF^2 seems not very effective in pixel-based RL. It would be much better if the authors discuss the potential ineffective scenarios. Thank you, and I'm open to change my score if the authors can further address these questions.

---

> > > ### Author Response · Authors · 2025-11-24
> > >
> > > We sincerely thank the reviewer for appreciating the additional experiments. We would like to clarify why our method is currently less effective in pixel-based environments and outline concrete next steps.
> > >
> > > 1. **Algorithmic mismatch with the original submission.**
> > > In the MinAtar environments, we adopted an on-policy actor–critic (AC) algorithm, while the primary experiments in the main paper were based on an off-policy approach. We made this choice because, for discrete action spaces, top-performing algorithms are typically either on-policy policy-gradient methods or off-policy Q-learning approaches.
> > > To stay as consistent as possible with the algorithm used in the original paper, we selected AC because it provides separate policy and value networks. However, unlike our original setup, the policy gradient here does not need to backpropagate directly through the value function. This change likely reduces the advantages of our approach relative to the baseline under the current configuration.
> > >
> > > 2. **Representing velocity fields from pixels is inherently difficult.** When observations are raw images, it is challenging to factorize the velocity field into matrices and vectors because the flattened representations are extremely high-dimensional. In the supplementary study we therefore tried two approaches. First, we restricted our operators to act only on the channel dimension, which inevitably discards information. Second, we trained an auxiliary autoencoder jointly with the policy to obtain a lower-dimensional latent space. While this helps in principle, the simultaneous training introduces instability for the generative model. In realistic pixel-based environments, one could instead use a pretrained VAE to obtain a stable mapping; training everything from scratch, as we did, increases the difficulty and likely harms performance.
> > >
> > > 3. **Current pixel-based experiments are still preliminary.** The present results should be viewed as an initial validation that our method can, in principle, extend to pixel-based settings, but realizing its full potential will require additional engineering and experimentation.
> > >
> > >
> > > Overall, we agree with the reviewer that extending the method to pixel-based settings is highly valuable. We will emphasize this direction in future work and plan to investigate pretrained representation learning pipelines that decouple the latent space from the policy optimization. Please let us know if there are additional questions; we are happy to elaborate. If you feel that our clarifications resolve your concerns, we would be grateful if you could consider raising the score. Thank you again for your thoughtful feedback and suggestions.

---

> > > > ### Comment · Reviewer_Cb48 · 2025-11-25
> > > >
> > > > Thanks for further explaination. I update my score to 6.

---

### Official Review · Reviewer_77u8 · 2025-11-01

**Soundness:** 3
**Presentation:** 2
**Contribution:** 2
**Rating:** 6
**Confidence:** 3

**Summary:**

This paper proposes Successor Flow Features ($SF^2$), which uses flow-matching technique to approximate Successor Measure (SM). The authors design a linear decomposition of the vector field $u$, separating the time-conditioned and time-invariant components. The time-invariant part is viewed as the learned representation of state action pair and is used by downstream reinforcement learning algorithms. Experiments are conducted on DeepMind Control Suite, and the results demonstrate the superiority of the proposed algorithm.

**Strengths:**

1.	The linear decomposition of $u$ is the key innovation of this paper and provides a clear architectural insight. It is well-motivated, and both its efficiency and sufficiency are thoroughly discussed.
2.	The proposed method can be used as a plug-in module and can be integrated in a wide range of RL algorithms.
3.	The experiments are clear and comprehensive, covering effectiveness, hyperparameter sensitivity and efficiency. Experimental details are clearly organized in appendices, making them easy enough to be reproduced.
4.	Experimental results clearly demonstrate the superiority of the proposed algorithm over the baseline algorithms, which are strong enough.

**Weaknesses:**

1.	The presentation of this paper is not sufficiently clear (see Questions for details)
2.	The literature review is somewhat limited. For instance, the paper does not cite Agarwal et al. [1], “Proto Successor Measure: Representing the Behavior Space of an RL Agent” (arXiv:2411.19418, 2024), which appears to be highly relevant to the topic and should be discussed for completeness.

**Questions:**

1.	Some of the notations used in this paper are not properly introduced.

    a)	The notation of Successor Measure introduced in lines 135-136 is defined as a probability distribution over state sets, but it is used later as a probability distribution over states (lines 170-171).

    b)	The notation of $ODE(\cdot, \cdot)$ (line 187) is not clearly introduced as a function, although the ODE used by flow matching is described in line 104.
2.	It would be better to provide a complete version of RL training pseudocode in Algorithm 1. The current version includes only value loss, but omits policy loss, which is confusing.
3.	There are existing works that have employed successor measures for RL. The author is encouraged to emphasize the importance of integrating successor measures for online RL representation learning.
4.	The learned representation is only used as input to $Q$ function. Is there a way to allow the policy $\pi$ to also leverage the learnt representation?

---

> ### Author Response · Authors · 2025-11-21
>
> Dear Reviewer 77u8,
>
> Thank you for taking the time to carefully review our paper. Please see our response and our revised manuscript to see the main changes we made to improve the manuscript and our responses to questions.
>
> > The notation of Successor Measure introduced in lines 135-136 is defined as a probability distribution over state sets, but it is used later as a probability distribution over states (lines 170-171).
>
> In the paper, we follow the measure-theoretic definition: the Successor Measure is a (discounted) measure on the measurable space, i.e., a set function $A\rightarrow R$. In later sections, for implementation, we work with a generative model that samples one state at a time. This corresponds to using the singleton events $A=\{s\},s\in S$. Thus, expressions like “distribution over states” are shorthand for the density/mass. No change of object is intended; we pass from the measure to its pointwise representation (density or mass) for practical modeling.
>
> > The notation of $ODE$ (line 187) is not clearly introduced as a function, although the ODE used by flow matching is described in line 104.
>
> Thanks for your advice, we will replace $ODE$ with $Euler$ to specify that we use the Euler method to solve the ODE.
>
> > It would be better to provide a complete version of RL training pseudocode in Algorithm 1. The current version includes only value loss, but omits policy loss, which is confusing.
>
> We put the representation loss part in the main part due to the page limit and provided the complete version of RL training pseudocode in the Appendix (Algorithm 2 and Algorithm 3) in the original submission.
>
> > There are existing works that have employed successor measures for RL. The author is encouraged to emphasize the importance of integrating successor measures for online RL representation learning.
>
> We have revised the Related Work section to include related works of using the SM method in a zero-shot RL setting to show that our approach is a complement to SM-related methods.
>
> >The learned representation is only used as input to $Q$
>  function. Is there a way to allow the policy to also leverage the learnt representation?
>
> Our representation $\psi(s,a)$ is optimized primarily for policy evaluation (the Q-function), where Successor Measure structure is directly useful. Since the Successor Measure $M^\pi$ is policy-dependent, coupling the actor to the same representation during online learning increases non-stationarity, causing the representation and the policy to chase a moving target, degrading stability and performance.
>
> We implemented the reviewer’s suggestion by feeding the representation into the policy: we modified the SM pathway to produce $\psi(s)$(state-only) and used $\pi_\theta(\psi(s))$with stop-gradient. However, the policy collapsed very quickly, and we think further exploring how to integrate the representation for policy is valuable, and leave it as future research.

---

> ### Author Response · Authors · 2025-11-27
>
> Dear reviewer 77u8,
>
> I hope this message finds you well. As the discussion period is approaching its deadline, we would like to kindly ask whether our rebuttal has addressed your concerns satisfactorily.If there are any additional points or clarifications you would like us to provide, please feel free to let us know. Your insights are invaluable to us, and we're eager to address any remaining issues to improve our work.
>
> Thank you for your careful review.
>
> Best,
>
> The Authors

---

### Author Response · Authors · 2025-12-03
**Rebuttal and discussion summary**

Dear ACs and reviewers,

We sincerely thank you for your time and thoughtful feedback.
The insightful suggestions and constructive feedback from the reviewers were crucial in guiding the improvement and overall enhancement of our work. These contributions have directly resulted in a clearer and more robust revised manuscript.
We would like to present a brief summary of the key points from the reviewer discussion and outline the improvements made to our manuscript since the original submission.

**Summary of our work:** We propose Successor Flow Features (SF$^2$), a representation-learning framework that uses flow matching to approximate successor measures (SM) and then linearly factorizes the learned vector field into a time-invariant embedding $\psi\in R^n$ and a time-dependent projection $\zeta \in R^{n\times \text{dim}_S}$. These features are plugged into off-policy algorithms (TD3/SAC) via the critic, aiming to improve sample efficiency on continuous control tasks.

**Addressing concerns about original experiments:** most reviewers considered our experimental evaluation to be sufficiently clear and convincing. In particular, Reviewer 77u8 noted that our experiments clearly demonstrate the superiority of the proposed algorithm over strong baseline methods; Reviewer Cb48 highlighted the consistent performance gains over standard baselines; and Reviewer ydVH commented that our method outperforms or matches baselines with clearly reported mean and standard deviation.
Reviewer kuve expressed significant concerns about the statistical significance of our reported improvements and noted the lack of learning curves. The reviewer assigned a very low score(2) to reflect this concern and recommended that we adhere to the evaluation protocol outlined in "Empirical Design in Reinforcement Learning." In the revised version, we followed the instructions from the reference paper and reviewers. We

* Expanded $5\rightarrow 15$ seeds across all DMC tasks.
* Added learning curves, AUC(0–T) metrics, IQM score, and confidence intervals following best practices. Our method still outperforms other baselines.
* Explain the original experiment setting, which followed the default setting and included 1M-step experiments showing that SF$^2$ maintains improvements even at shorter horizons.
* Included experiments with different gradient updates per step to describe how our method changed under different training steps as an extra supplement orthogonal to different environment steps.

Due to the short discussion period, Reviewer kuve didn't give feedback on our rebuttal. We believe these additions collectively can demonstrate the robustness of our method and effectively alleviate concerns regarding statistical significance and potential bias in selecting experimental configurations.

**Addressing concerns regarding baselines and additional settings:** to address Reviewer Cb48 and Reviewer ydVH’s concerns about missing baselines and settings, we carefully implemented
* SPR, adapted for state-based settings with noise augmentation (including augmentation sweeps).
* Added MinAtar experiments (discrete actions + pixel-like observations) with two SF$^2$ variants (channel-interpolated, AE-based).
* Added robustness experiments with structured observation noise.
* Conducted an additional experiment where we augmented a representative CRL method with our auxiliary representation task with anonymous code link
* Conducted an additional experiment on the Flow-based policy with an anonymous code link

We believe our additional experiments solve Reviewer Cb48's concern, as Reviewer Cb48 appreciated the detailed supplementary experiments and further clarification in our rebuttal, and raised the score from 4 to 6(on Nov. 25th), noting that the concerns were fully resolved. And due to the short period, Reviewer ydVH didn't give further response on our additional experiments.

**Addressing concerns about clarity, writing, and insight:** we edited the related work part and main text(response to Reviewer 77u8) added visualizations of $\psi,\zeta$ and the flow field $u$(response to Reviewer ydVH), softened claims(response to Reviewer kuve), provided additional intuition on why aligning vector fields (response to Reviewer ydVH) and clarified insight into the novelty of learning the SM(response to Reviewer kuve) and clarified that the added compute primarily benefits critic representation quality(response to Reviewer ydVH, Cb48, kuve).

We respectfully ask that the Area Chair recognize both the contributions of our paper and the substantial effort made to thoroughly address every reviewer's concern.
We believe the revised manuscript is significantly stronger: scientifically clearer, experimentally more rigorous, and better positioned within the literature.

Thank you for your time and consideration.

The Authors

---

### Meta-Review · Area_Chair_JRVZ · 2025-12-30

**Summary:**

1.Lack of discussions with related works, proposed by Reviewer 77u8.

2.Insufficient justification for successor representations in online RL, proposed by Reviewer Cb48.

3.Lack of important baselines, such as SPR (Self-Predictive Representations), proposed by Reviewer Cb48.

4.Experiments are only repeated for 5 trials and the paper reports large standard deviations, proposed by Reviewer kuve.

5.How the paper is distinct from prior works, proposed by Reviewer kuve.

6.Theoretical guarantee of the paper is limited, proposed by Reviewer ydVH.

7.Lack of results on problems with high-dimensional observation spaces, proposed by Reviewer ydVH.

8.Lack of comparisons with flow/diffusion-based RL methods, such as, DSLR, TDFlow, contrastive RL, proposed by Reviewer ydVH.

**Reviewer Concerns:**

1.Lack of discussions with related works, proposed by Reviewer 77u8: during the rebuttal phase, the authors revise the Related Work section to include related works of using the SM method in a zero-shot RL setting. Thus I think the point is addressed.

2.Insufficient justification for successor representations in online RL, proposed by Reviewer Cb48: during the rebuttal phase, the authors does not reply to it. Thus I think this point is still outstanding.

3.Lack of important baselines, such as SPR (Self-Predictive Representations), proposed by Reviewer Cb48: during the rebuttal phase, the authors compare with SPR and discuss the difference between SPR and SF^2. Results show that the proposed method outperforms SPR. Thus I think the point is addressed.

4.Experiments are only repeated for 5 trials and the paper reports large standard deviations, proposed by Reviewer kuve: during the rebuttal phase, the authors add results of 15 random seeds. Thus I think the point is addressed.

5.How the paper is distinct from prior works, proposed by Reviewer kuve: during the rebuttal phase, the authors discuss the difference between the main method with prior works. They argue that the main focus of the paper is to aid representation learning by SM, and to enhance reinforcement learning algorithms. Thus I think the point is addressed.

6.Theoretical guarantee of the paper is limited, proposed by Reviewer ydVH: during the rebuttal phase, the authors claim that Section 3.3 is just as intuition, not an algorithmic step. Thus I think the point is addressed.

7.Lack of results on problems with high-dimensional observation spaces, proposed by Reviewer ydVH: during the rebuttal phase, the authors claim they leave it as a future work. Thus I think this point is still outstanding.

8.Lack of comparisons with flow/diffusion-based RL methods, such as, DSLR, TDFlow, contrastive RL, proposed by Reviewer ydVH: during the rebuttal phase, the authors claim that these methods target different settings. Thus I think the point is addressed.

**Reviewer Scores:**

Reviewer 77u8 would keep his or her score as 6 if he or she has been able to participate fully in the discussion.

Reviewer Cb48 would improve his or her score from 4 to 6 if he or she has been able to participate fully in the discussion.

Reviewer kuve would improve his or her score from 2 to 4 if he or she has been able to participate fully in the discussion.

Reviewer ydVH would keep his or her score as 6 if he or she has been able to participate fully in the discussion.

---

### Decision · Program_Chairs · 2026-01-26

Accept (Poster)